# Infrared molecular fingerprinting of blood-based liquid biopsies for the detection of cancer

Marinus Huber[1,2†], Kosmas V Kepesidis[1,2†], Liudmila Voronina[1], Frank Fleischmann[1], Ernst Fill[2], Jacqueline Hermann[1], Ina Koch[3], Katrin Milger-Kneidinger[4], Thomas Kolben[5], Gerald B Schulz[6], Friedrich Jokisch[6], Jürgen Behr[4], Nadia Harbeck[5], Maximilian Reiser[7], Christian Stief[6], Ferenc Krausz[1,2], Mihaela Zigman[1,2*]

[1]Ludwig Maximilians University Munich (LMU), Department of Laser Physics, Garching, Germany; [2]Max Planck Institute of Quantum Optics (MPQ), Laboratory for Attosecond Physics, Garching, Germany; [3]Asklepios Biobank for Lung Diseases, Department of Thoracic Surgery, Member of the German Center for Lung Research, DZL, Asklepios Fachkliniken München-Gauting, Munich, Germany; [4]University Hospital of the Ludwig Maximilians University Munich (LMU), Department of Internal Medicine V, Munich, Germany; [5]University Hospital of the Ludwig Maximilians University Munich (LMU), Department of Obstetrics and Gynecology, Breast Center and Comprehensive Cancer Center (CCLMU), Munich, Germany; [6]University Hospital of the Ludwig Maximilians University Munich (LMU), Department of Urology, Munich, Germany; [7]University Hospital of the Ludwig Maximilians University Munich (LMU), Department of Clinical Radiology, Munich, Germany

*For correspondence: mihaela.zigman@mpq.mpg.de

†These authors contributed equally to this work

Competing interest: The authors declare that no competing interests exist.

**Abstract** Recent omics analyses of human biofluids provide opportunities to probe selected species of biomolecules for disease diagnostics. Fourier-transform infrared (FTIR) spectroscopy investigates the full repertoire of molecular species within a sample at once. Here, we present a multi-institutional study in which we analysed infrared fingerprints of plasma and serum samples from 1639 individuals with different solid tumours and carefully matched symptomatic and non-symptomatic reference individuals. Focusing on breast, bladder, prostate, and lung cancer, we find that infrared molecular fingerprinting is capable of detecting cancer: training a support vector machine algorithm allowed us to obtain binary classification performance in the range of 0.78–0.89 (area under the receiver operating characteristic curve [AUC]), with a clear correlation between AUC and tumour load. Intriguingly, we find that the spectral signatures differ between different cancer types. This study lays the foundation for high-throughput onco-IR-phenotyping of four common cancers, providing a cost-effective, complementary analytical tool for disease recognition.

## Introduction

To address the ever-growing cancer incidence and mortality rates, effective treatment methods are indispensable (*Bray et al., 2018*). They rely on detection of the disease at the earliest possible stage to allow antitumour interventions and thus improve survival rates (*Bannister and Broggio, 2016*; *Schiffman et al., 2015*). Early detection is therefore a crucial factor in the global fight against cancer.

However, the clinical benefits versus the potential harms and costs of several cancer detection approaches remain controversial (*Schiffman et al., 2015*). Due to the limited sensitivity and specificity of current medical diagnostics, cancer can either be overlooked (false negatives) or falsely detected

(false positives), leading to either delayed interventions or unnecessary, potentially harmful investigations or psychological stress (*Srivastava et al., 2019*). Hence, there is a high need to complement current medical diagnostics with time- and cost-efficient, non-invasive or minimally invasive methods that could possibly lead to new screening and detection approaches, prior to tissue-biopsy-based molecular profiling or prognosis (*Wan et al., 2017*).

Molecular analyses of human serum and plasma provide systemic molecular information and enabling novel routes of diagnostics (*Amelio et al., 2020*; *Wan et al., 2017*). So far, most liquid biopsies predominantly rely on the analysis of a few pre-selected analytes and biomarkers. Although the emergence of highly sensitive and molecule-specific methods in the fields of proteomics (*Geyer et al., 2019*; *Geyer et al., 2017*; *Uzozie and Aebersold, 2018*), metabolomics (*Roig et al., 2017*; *Xia et al., 2013*), and genomics (*Abbosh et al., 2017*; *Han et al., 2017*; *Otandault et al., 2019*) has led to the discovery of thousands of different biomarker candidates, only a few of them have been validated and transferred to the clinic so far (*Poste, 2011*).

Technological developments of the last decade brought a paradigm change regarding liquid biopsies. Instead of relying on a single molecular marker, recent approaches focus on combining information across a broad range of molecules to investigate changes in molecular patterns and identify disease-specific physiologies. However, the combination of various omics techniques (i.e. multi-omics) still requires complex and target-specific sample preparation as well as elaborate ways of merging different datasets (*Hasin et al., 2017*; *Karczewski and Snyder, 2018*; *Malone et al., 2020*; *Yoo et al., 2018*). Moreover, increasing the number of analytical methods involved often leads to unfeasibly high costs for broad clinical use.

This is where infrared molecular spectroscopy prevails – it captures signals from all types of molecules in a sample in a single time- and cost-effective measurement in a label-free fashion. When applied to blood serum or plasma samples, infrared spectroscopy provides a so-called infrared molecular fingerprint (IMF) reflecting the chemical composition of a sample, that is, the person's molecular blood phenotype (*Huber et al., 2021*). Even though the IMF of a highly complex biofluid such as blood serum and plasma can only partially be traced back to its molecular origin, it may deliver a plethora of information sensitive and specific to the health state of the individual. In a recent longitudinal study, we have shown that defined workflows to collect, store, process, and measure human liquid biopsies lead to reproducible IMFs in healthy, non-symptomatic individuals that are stable over clinically relevant time scales (*Huber et al., 2021*).

Numerous studies have shown the potential of blood-based IMFs for the detection of cancer, notably brain (*Butler et al., 2019*; *Hands, 2014*; *Sala et al., 2020b*), breast (*Backhaus et al., 2010*; *Elmi et al., 2017*; *Ghimire et al., 2020*; *Zelig et al., 2015*), bladder (*Ollesch et al., 2014*), lung (*Ollesch et al., 2016*), prostate (*Medipally et al., 2020*), and other cancer entities (*Anderson et al., 2020*; *Ollesch et al., 2014*; *Sala et al., 2020a*), with some of the studies reporting specificities and sensitivities higher than 90% (*Anderson et al., 2020*; *Backhaus et al., 2010*; *Butler et al., 2019*; *Ghimire et al., 2020*; *Medipally et al., 2020*; *Ollesch et al., 2014*). Despite these promising initial results, only a few studies involved more than 75 individuals per group (*Anderson et al., 2020*). Additionally, the majority of these studies had a high risk of bias due to patient selection applied (*Anderson et al., 2020*). In fact, it was shown that IMFs are susceptible to external confounding factors, such as those related to sample handling and data collection, as well as to inherent biological variations (e.g. age and gender) unrelated to cancer (*Diem, 2018*). Furthermore, the differences observed in IMFs may be due to the innate immune response and other concomitant factors (*Diem, 2018*; *Fabian et al., 2005*). Thus, the specificity of IMF to a certain cancer must be evaluated by investigation of appropriate, carefully selected reference groups.

Altogether, there is a need for studies that address the issues listed above by (i) systematically investigating the pre-analytical factors (*Cameron et al., 2020*; *Huber et al., 2021*), (ii) studying the molecular origin of the infrared fingerprints (*Voronina et al., 2021*), and (iii) adequately applying machine learning tools with involvement of a sufficient number of participants. To date, the latter requirement has only been met in studies investigating the applicability of infrared fingerprinting to bladder (*Ollesch et al., 2014*), breast (*Backhaus et al., 2010*), and brain cancer detection (*Butler et al., 2019*). In addition to the capacity to detect cancer, whether different cancer entities have sufficiently different infrared spectral signatures to be distinguished from each other has so far not been evaluated.

Our present multi-institutional, multi-disease study addresses the above issues to rigorously assess the feasibility of IMFs for high-throughput detection of four common cancer entities as phenotypes, thus referred to as '*onco-IR-phenotyping*.' Using Fourier-transform infrared (FTIR) transmission spectroscopy of liquid samples, we measured blood serum and plasma samples from 1927 individuals, among these 161 breast cancer, 118 bladder cancer, 278 prostate cancer, and 214 lung cancer patients, prior to any cancer-related therapy, along with non-symptomatic reference individuals and study participants with diseases and/or benign pathologies of the same organ (i.e. organ-specific symptomatic references). By applying support vector machine (SVM) to train models for binary classification, we obtained detection efficiencies in the range of 0.78–0.89 (area under the receiver operating characteristic [ROC] curve [AUC]), with the detection efficiency strongly correlating with the severity of the disease. The results of this prospectively conducted study suggest that infrared fingerprinting of liquid plasma and serum may offer a means of robust and reliable detection of different types of cancer. Furthermore, we reveal that the spectral signatures attributable to different cancer types differ significantly from each other, which facilitates classification between different states and thus carries a translational potential not previously reported.

## Results

### Study setup and workflow

In this study, we tested infrared molecular spectroscopy for medically relevant blood profiling in a prototypical multi-institutional setting, assessing the usefulness of IMFs as a source of complementary information for cancer diagnostics. The study included cohorts of therapy-naïve, lung, prostate, bladder, and breast cancer patients (cases), and organ-specific symptomatic references as well as non-symptomatic reference individuals (*Figure 1a*, *Figure 1—source data 1*).

Blood sera and plasma were collected at several clinical sites according to well-defined standard operating procedures to minimize pre-analytical errors (*Figure 1b*; *Huber et al., 2021*). An automated sample delivery system was applied for high-throughput, high-reproducibility, and cost-efficient infrared fingerprinting of liquid sera and plasma of 1927 individuals with an FTIR spectrometer (*Figure 1c*). Special care was taken to match the characteristics of the case and reference cohorts for each question separately – by age, gender, and body mass index (BMI) – to avoid patient selection bias, although this step reduced the number of individuals analysed within this study to 1639 (*Figure 1d*). The acquired IMFs were used for training machine learning models to perform binary classification of the samples (*Figure 1e*) into case and reference groups, allowing the investigation of various clinically relevant questions (see below). Model training was performed by applying SVM algorithm to pre-processed IMFs by splitting the data into train and test sets, employing 10-fold cross-validation, repeated 10-times with randomization. For assessing the classification performance, we evaluated the AUC of the respective ROC curves for the test sets.

### Diagnostic performance of infrared molecular fingerprinting for cancer detection

In a first step, we evaluated the diagnostic performance of IMFs obtained from serum samples for the binary classification of each of the four common cancer types individually against matched non-symptomatic reference groups (see *Table 1* and *Figure 2—source data 1* for details on the characteristics of the individual cohorts). Since our approach produces results in terms of continuous variables (disease likelihood) rather than binary outcomes (disease, non-disease), we use the AUC of the ROC as the main performance metric, and thus take advantage of incorporating information across multiple operating points, not limited to a particular clinical scenario.

The highest detection efficiencies in the test sets were obtained for the lung and breast cancer cohort SVM models, with a ROC AUC of 0.89 and 0.88, respectively (*Figure 2a*). A lower classification performance of 0.79 and 0.78 (ROC AUC) was obtained for the prostate and bladder cancer cohorts, respectively. *Table 1* also lists the optimal combination (see Methods) of sensitivity and specificity for all cancer entities. For making our results comparable to other studies and possibly to gold standards in cancer detection, we present lists with sensitivity/specificity pairs (see *Table 1*). In particular, we present the optimal pairs extracted by minimizing the distance between the ROC curve and the

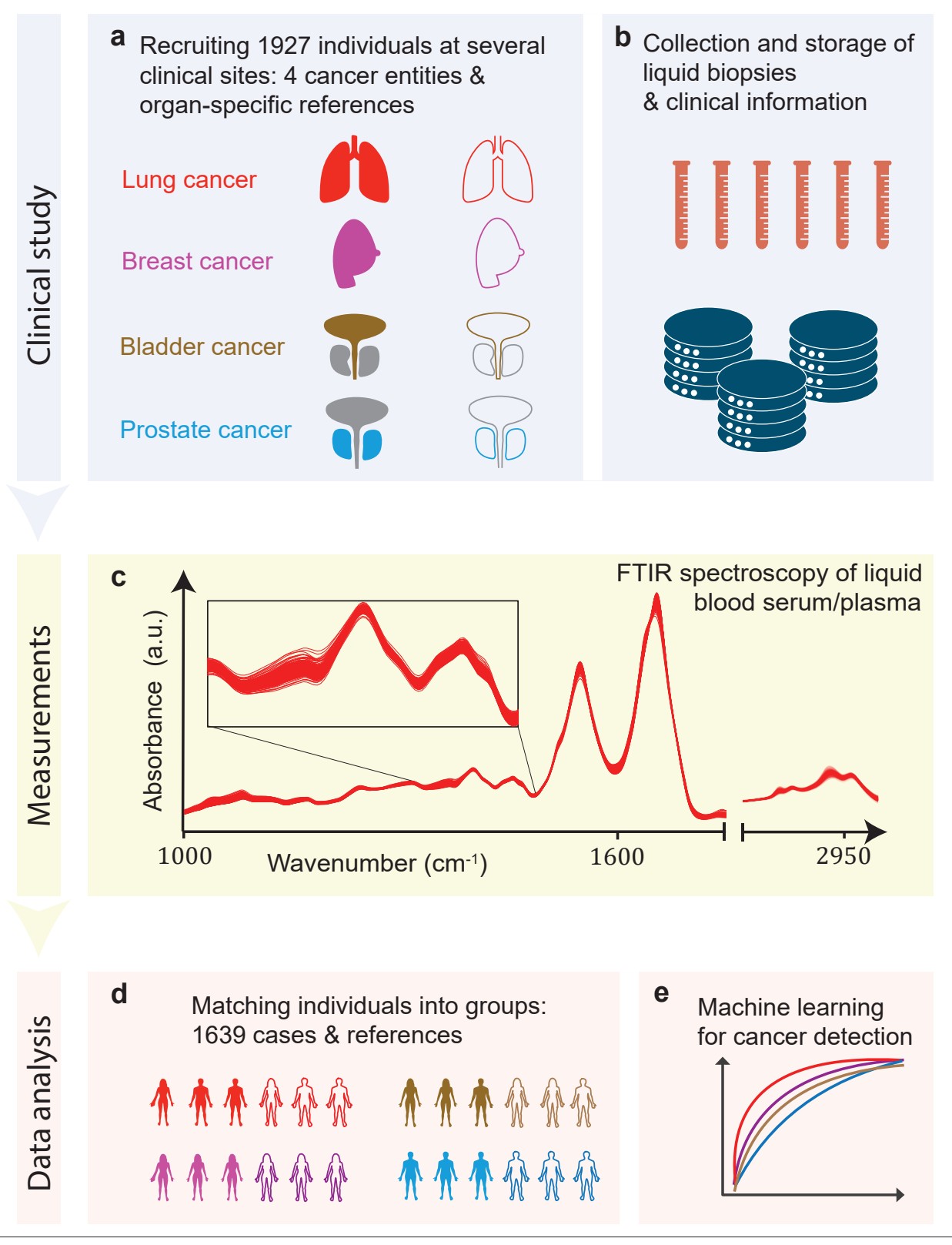

**Figure 1.** Infrared molecular fingerprinting workflow and clinical study design. (**a**) Cohorts of therapy-naïve, lung, breast, prostate, and bladder cancer patients (cases), and organ-specific symptomatic references as well as non-symptomatic reference individuals were recruited at three different clinical sites – in total, 1927 individuals. (**b**) Blood samples from all individuals were drawn, and sera and plasma were prepared according to well-defined standard operating procedures. (**c**) Automated Fourier-transform infrared spectroscopy of liquid bulk sera and plasma were used to obtain IMFs. The

*Figure 1 continued on next page*

*Figure 1 continued*

displayed IMFs were pre-processed using water correction and normalization (see Methods). (**d**) For each clinical question studied, the characteristics of the case and the reference cohorts were matched for age, gender, and body mass index (BMI) to avoid patient selection bias. This resulted in total number of 1639 individuals upon matching. (**e**) Machine learning models were built on training datasets and evaluated on test datasets to separately evaluate the efficiency of classification for each of the four cancer entities.

The online version of this article includes the following figure supplement(s) for figure 1:

**Source data 1.** Breakdown of the overall participant pool used within the study.

upper-left corner – a standard practice in studies of this type. In addition, we set the specificity to 95% and present the resulting sensitivities.

In clinical practice, however, patients may suffer from pathologies that affect the same organ as the cancer under scrutiny. Therefore – in a second step, we tested the capability of IMFs to classify cancer, when organ-specific comorbidities (e.g. chronic obstructive pulmonary disease [COPD] in the lung cancer cohort) and organ-specific benign conditions (e.g. hamartoma of the lung in the lung cancer cohort or benign prostate hyperplasia [BPH] in the prostate cancer cohort – see *Figure 1—source data 1* for details) were added to the reference group. In this case, the detection efficiency decreased significantly, from 0.89 to 0.77, for lung cancer and slightly, from 0.79 to 0.75, for prostate cancer (*Figure 2b*). If the reference group contained only organ-specific symptomatic references, the detection efficiency was reduced further, to 0.74 for lung cancer and 0.70 for prostate cancer (*Figure 2c*).

To test whether sample collection, handling, and storage have a potential influence on classification results, we examined data from matched, non-symptomatic, healthy individuals from the three major clinics using principal component analysis (PCA). Considering the first five principal components (responsible for 95% of the explained variance), we could not observe any clustering effect related to data from different clinics (*Figure 2—figure supplement 1*). However, potential bias due to the above-mentioned influences cannot be fully excluded at the present stage. To this end, samples at different clinical sites are being collected to form a large independent test dataset, specifically designed to allow us evaluate the effects of clinical covariates – as well as measurement-related ones – relevant for the proposed IMF-based medical assay. One typically obtains a different AUC by using different control groups, collected at different sites (*Figure 2—source data 3*). These variations have many potential causes, including measurement-related effects, differences in sample handling, unobserved differences between the clinical populations recruited at different clinical sites, and of course the size of the training sets used for model training, which can significantly affect the model performance.

**Table 1.** Detection efficiency for different binary classifications.

Different cancer types were compared to each other, as well as the impact of using different reference groups was analysed. Detailed cohort characteristics can be found in Figure 2—source data 1 (NSR: non-symptomatic references; MR: mixed references; SR: symptomatic references; AUC: area under the receiver operating characteristic curve; *sensitivity and specificity values are obtained by minimizing the distance of the receiver operating characteristic [ROC] curve to the upper-left corner).

| Clinical question for binary classification | # of Individuals | AUC | Sensitivity/ specificity* | sensitivity at 95% specificity |
|---|---|---|---|---|
| Lung cancer vs. NSR | 214/193 | 0.89 ± 0.05 | 0.86/0.79 | 0.45 |
| Lung cancer vs. MR | 214/208 | 0.77 ± 0.06 | 0.72/0.67 | 0.36 |
| Lung cancer vs. SR | 214/143 | 0.74 ± 0.07 | 0.67/0.71 | 0.24 |
| Prostate cancer vs. NSR | 278/278 | 0.78 ± 0.06 | 0.71/0.71 | 0.36 |
| Prostate cancer vs. MR | 278/278 | 0.75 ± 0.06 | 0.71/0.68 | 0.23 |
| Prostate cancer vs. SR | 278/278 | 0.70 ± 0.06 | 0.65/0.68 | 0.20 |
| Breast cancer vs. NSR | 161/161 | 0.88 ± 0.06 | 0.82/0.81 | 0.35 |
| Bladder cancer vs. NSR | 118/118 | 0.79 ± 0.09 | 0.72/0.73 | 0.23 |

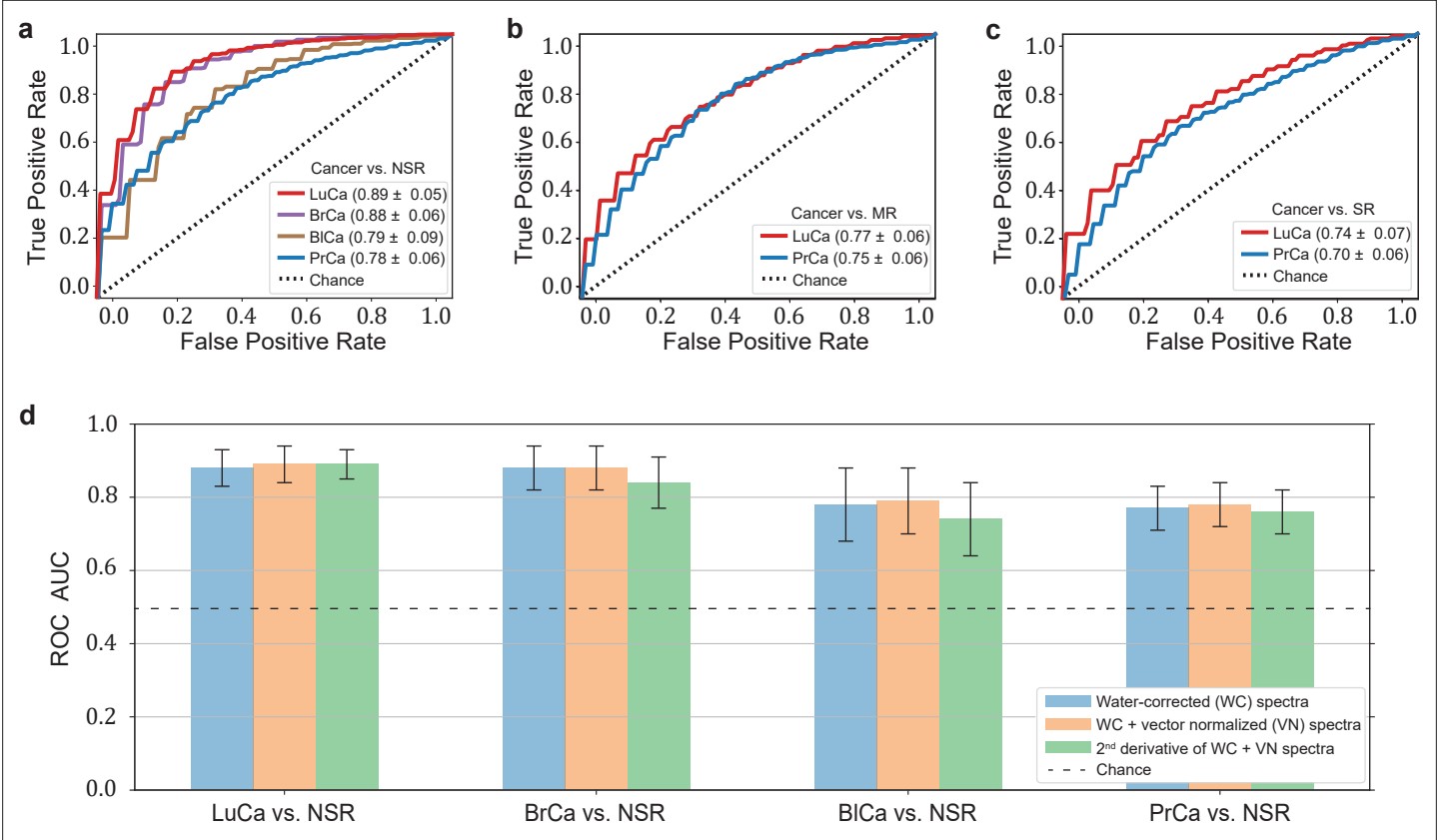

**Figure 2.** Diagnostic performance of lung, prostate, bladder, and breast cancer detection based on infrared molecular fingerprints (IMFs) of blood sera. Receiver operating characteristic (ROC) curves for the binary classification of the test set with support vector machine (SVM) models trained on water-corrected and vector-normalized IMFs. The different cancer entities were tested against (**a**) non-symptomatic references, (**b**) mixed references that also include organ-specific symptomatic references, and (**c**) organ-specific symptomatic references only. Detailed cohort characteristics can be found in *Figure 2—source data 1*. (**d**) Area under the receiver operating characteristic curve (AUC) for the test sets according to different spectral pre-processing of the IMFs. The error bars show the standard deviation of the individual results of the cross-validation (LuCa: lung cancer; PrCa: prostate cancer; BrCa: breast cancer; BlCa: bladder cancer; NSR: non-symptomatic references; MR: mixed references; SR: symptomatic references).

The online version of this article includes the following source data and figure supplement(s) for figure 2:

**Source data 1.** Characteristics of the matched groups of individuals utilized for the analysis as presented in Table 1, *Figures 2 and 3a-c*.

**Source data 2.** Zipped folder with trained machine learning models and application instructions.

**Source data 3.** Potential impact of clinical site to classification performance.

**Figure supplement 1.** Unsupervised comparison between data from the three clinical sites as well as quality control (QC) analysis of measurements.

**Figure supplement 1—source data 1.** Characteristics of the matched groups utilized for the analysis presented in *Figure 2—figure supplement 1*.

**Figure supplement 2.** Performance comparison of serum- and plasma-based fingerprints for cancer detection.

**Figure supplement 2—source data 1.** Characteristics of the matched groups utilized for the analysis presented in *Figure 2—figure supplement 2*.

Although important, it is currently not feasible to rigorously disentangle these effects. Furthermore, we investigated the influence of different pre-processing of the IMFs on the classification results and reassuringly found that these are not significantly affected by the applied pre-processing (*Figure 2d*). Model diagnostics yielded no signs of overfitting as we added different layers of pre-processing into the pipeline (see Methods for details). Since water-corrected and vector-normalized spectra typically resulted in slightly higher AUCs but still low overfitting, this pre-processing was kept in all other analyses.

It is generally known that blood serum and blood plasma provide largely overlapping molecular information, and both can be used as a basis for many further investigations. The extent to which this also applies to infrared fingerprinting has not been extensively studied. In a previous comparative

study, we were able to show that healthy phenotypes can be better identified on the basis of serum (*Huber et al., 2021*).

Here we compare the diagnostic performance of IMFs from serum and plasma collected from the same individuals for the detection of lung and prostate cancer compared to non-symptomatic and symptomatic organ-specific references. Given that plasma samples were only available for a subset of the lung and prostate cohorts, the results for serum slightly deviate from those presented above due to the different cohort characteristics (*Figure 2—figure supplement 2—source data 1*). The detection efficiency based on IMFs from plasma samples was 3% higher in the case of lung cancer and 2% higher in the case of prostate cancer than the same analysis based on IMFs from serum samples. In both cases, the difference in AUC was only of low significance. It is noteworthy that the corresponding ROC curves show similar behaviour (*Figure 2—figure supplement 2*). These results suggest that either plasma or serum samples could in principle be used for detection of these cancer conditions. However, for carefully assessing whether (i) the same amount of information is contained in both biofluids and (ii) whether this information is encoded in a similar way across the entire spectra requires yet an additional dedicated study with higher sample numbers.

## Investigation of cancer-specific infrared signatures

In many clinical settings, a simple binary classification may not be sufficient; instead, a simple, quick, and reliable test that indicates a specific cancer or disease is preferred. To investigate the possible existence of cancer-specific IMFs (or onco-IR-phenotypes), we first examined and compared the spectral signatures that are relevant for distinguishing cancer cases from non-symptomatic references. For this purpose, we evaluated the differential fingerprints (defined as the difference between the mean IMF of the case cohort and that of the reference cohort), determined the two-tailed p-value of Student's t-test, and calculated the AUC per wavenumber using the U statistic of a Mann–Whitney U test (see Methods) for all cohorts (*Figure 3a–c*). The obtained patterns differed significantly for all four cancer entities.

It is noteworthy that for lung and breast cancer the magnitude of the differential fingerprint compared to the variation of the IMFs of the reference group (grey area in *Figure 3a*) is more pronounced than for bladder and prostate cancer. This is also reflected in the p-values (*Figure 3b*), reaching many orders of magnitude lower levels for the former cancer entities, and higher spectrally resolved AUCs (*Figure 3c*). Compared to evaluation based on the entire spectral range, spectral containment significantly reduces detection efficiency for all cancer entities, although the reduction is smaller for lung and breast cancer. For these two cancer entities, classification based on a few selected spectral regions is possible. By contrast, for prostate and bladder cancer, the cancer-relevant information appears to be distributed over the entire spectral range and a high classification rate relies on the entire spectral range accessible.

The fact that the cancer entities studied here have different spectral signatures raises the question of whether it is possible to get first indication of the type of cancer detected, which can become relevant, for example, if the primary origin of a cancer type is unknown. Therefore, we performed a multiclass classification aiming to distinguish between lung, bladder, and breast cancer for a matched female cohort (*Figure 3d*) and between lung, bladder, and prostate cancer for a matched male cohort (*Figure 3e*). Note that the number of included cancer cases had to be significantly reduced in multiclass classification, as compared to the binary classification, in order to preserve balanced cohort characteristics. Details on this are given in *Figure 3—source data 1*. Overall, the classification accuracy was 73% and 74%, respectively. These findings suggest that primary tumours evolving in different organs indeed induce differing changes in the overall molecular composition of blood sera – as reflected in differing spectral signatures – and thus offering potential for cancer stratification in future. However, due to the small dataset, these findings need to be verified with larger, independent cohorts.

Often, a patient may reveal symptoms suggestive of a certain cancer entity (e.g. lung cancer), but same time also symptoms indicative of additional further diseases or benign conditions. Therefore, we tested the ability of infrared fingerprinting to detect signatures that would be specific to lung and prostate cancer in comparison to organ-specific (benign) diseases in each case. To this end, we evaluated the differential fingerprints of the different organ-specific symptomatic references compared to non-symptomatic individuals and compared these signatures to the cancer-related IMFs, respectively (*Figure 3—figure supplement 1*). We found that the differential fingerprint for asthma and lung

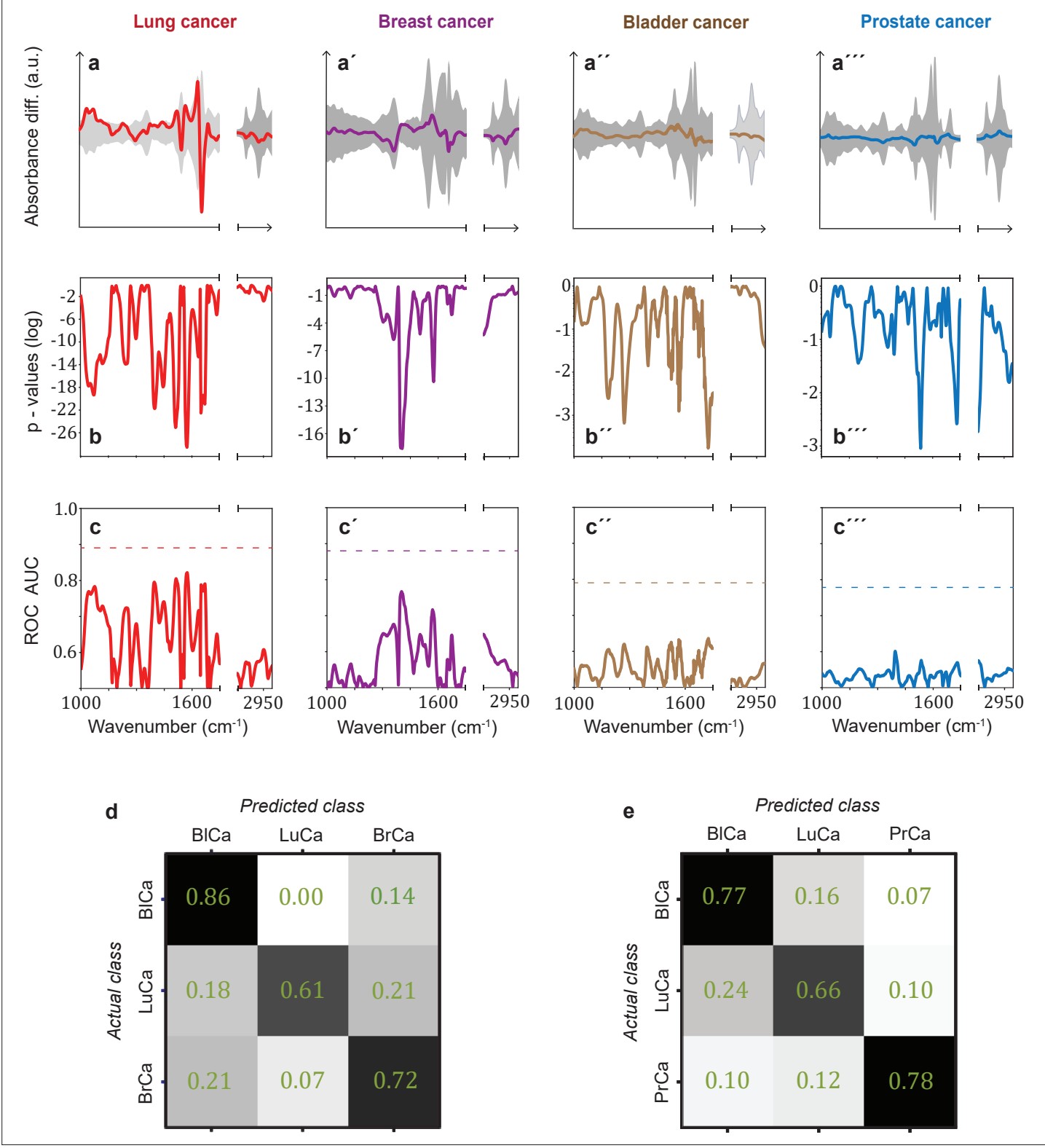

**Figure 3.** Infrared spectral signatures of lung, prostate, bladder, and breast cancer. (**a-a'''**) Differential fingerprints (standard deviations of the reference cohorts are displayed as grey areas), (**b-b'''**) two-tailed p-value of Student's t-test, and (**c-c'''**) area under the receiver operating characteristic curve (AUC) per wavenumber (extracted by application of Mann–Whitney U test) compared to the AUC of the combined model (dashed horizontal lines). Confusion matrix summarizing the per-class accuracies of multiclass classification of (**d**) lung, bladder, and breast cancer (matched female cohort) with overall model accuracy of 0.73 ± 0.11, and (**e**) lung, bladder, and prostate cancer (matched male cohort) with overall model accuracy of 0.74 ± 0.13.

*Figure 3 continued on next page*

*Figure 3 continued*

Detailed cohort characteristics can be found in ***Figure 3—source data 1***. Chance level for the three-class classification corresponds to 0.33 (LuCa: lung cancer; PrCa: prostate cancer; BrCa: breast cancer; BlCa: bladder cancer).

The online version of this article includes the following source data and figure supplement(s) for figure 3:

**Source data 1.** Characteristics of the matched groups utilized for the analysis presented in ***Figure 3d and e***.

**Figure supplement 1.** Comparison of signatures from different organ-specific pathologies.

**Figure supplement 1—source data 1.** Characteristics of the matched groups utilized for the analysis presented in ***Figure 3—figure supplement 1***.

hamartoma clearly differs from the ones obtained for lung cancer and COPD. However, the differential fingerprints of the latter two diseases, although distinguishable, exhibit strong similarities in their main spectral features. This explains why the presence of COPD in the reference group lowers the detection efficiency of lung cancer (***Figure 2a*** vs. ***Figure 2b***). In contrast, the differential fingerprints of BPH and prostate cancer differ considerably. Consequently, BPH in the reference group does not strongly affect the detection efficacy of prostate cancer.

Lung cancer is often accompanied by COPD, and the previous analysis showed that the differential fingerprint of COPD and lung cancer exhibits similarities. Thus, we investigated whether infrared fingerprinting could possibly identify any infrared signals specific only to lung cancer (and not to COPD). Towards that end, we separated individuals from the above analysis into sub-cohorts with subjects negative and positive for COPD. We found that the detection of lung cancer was less efficient when the reference cohort contained only COPD-positive individuals (***Figure 4—figure supplement 1***,). Both conditions (lung cancer and COPD) are often accompanied by an inflammatory response. Considering that the spectral signatures relevant for cancer detection are based on typical molecular changes that also occur in inflammatory conditions (***Voronina et al., 2021***), the presence of COPD likely masks, at least in part, cancer-relevant signals.

Another relevant question is whether a distinction between cancer and corresponding organ-specific benign pathologies can be made. Here, we evaluated to what extent this was possible for lung and prostate cancer. In both cases, we observed that the cancer detection was only moderately higher against a group of non-symptomatic individuals as compared to a group of patients with a benign condition (lung hamartoma and BPH, respectively; see ***Figure 4a and b***).

Finally, we explored the possibility of creating multiclass classification models to simultaneously discriminate between multiple groups: cancer patients, individuals with benign conditions, and non-symptomatic reference subjects (***Figure 4c and d***). In both cases, the classification accuracy was well above chance. Although the accuracy may not yet be sufficient for clinical use, these accuracies may significantly improve with more samples available for training.

## Dependence of cancer detection performance on tumour progression

Challenges for cancer detection include the enormous biological and clinical complexity of cancer, and detection is further complicated by the significant intratumour heterogeneity (***McGranahan and Swanton, 2017***) as well as by the impact of the tumour microenvironment (***Boothby and Rickert, 2017***). To evaluate whether the blood-based IMFs are sensitive to tumour progression, we first investigated whether the binary classification efficiency depends on tumour size, characterized in terms of clinical TNM staging (***Amin et al., 2017***).

In general, we observe that the classification efficiency exhibits a positive correlation with tumour size or tumour grade. In the case of lung cancer, when compared to the non-symptomatic references, the classification efficiency for T4 tumours is (in terms of AUC) 9% higher than that for T1 tumours (***Figure 5a***). Also, for breast and bladder cancer, a significantly higher detection efficiency for T3 tumours was observed. This is also reflected by the fact that a more pronounced differential fingerprint can be found in these cancers in higher T classes (***Figure 5a–c***). Although the absolute (integrated) deviation – between the cases and the matched references – increases for all four cancer phenotypes, the spectral features are partly different for the different T stages. This could be due to the fact that, due to the moderate number of individuals considered, the actual onco-IR-phenotype is masked by biological variability, or that the heterogeneity of tumour growth leads to different molecular changes and thus to different IMFs.

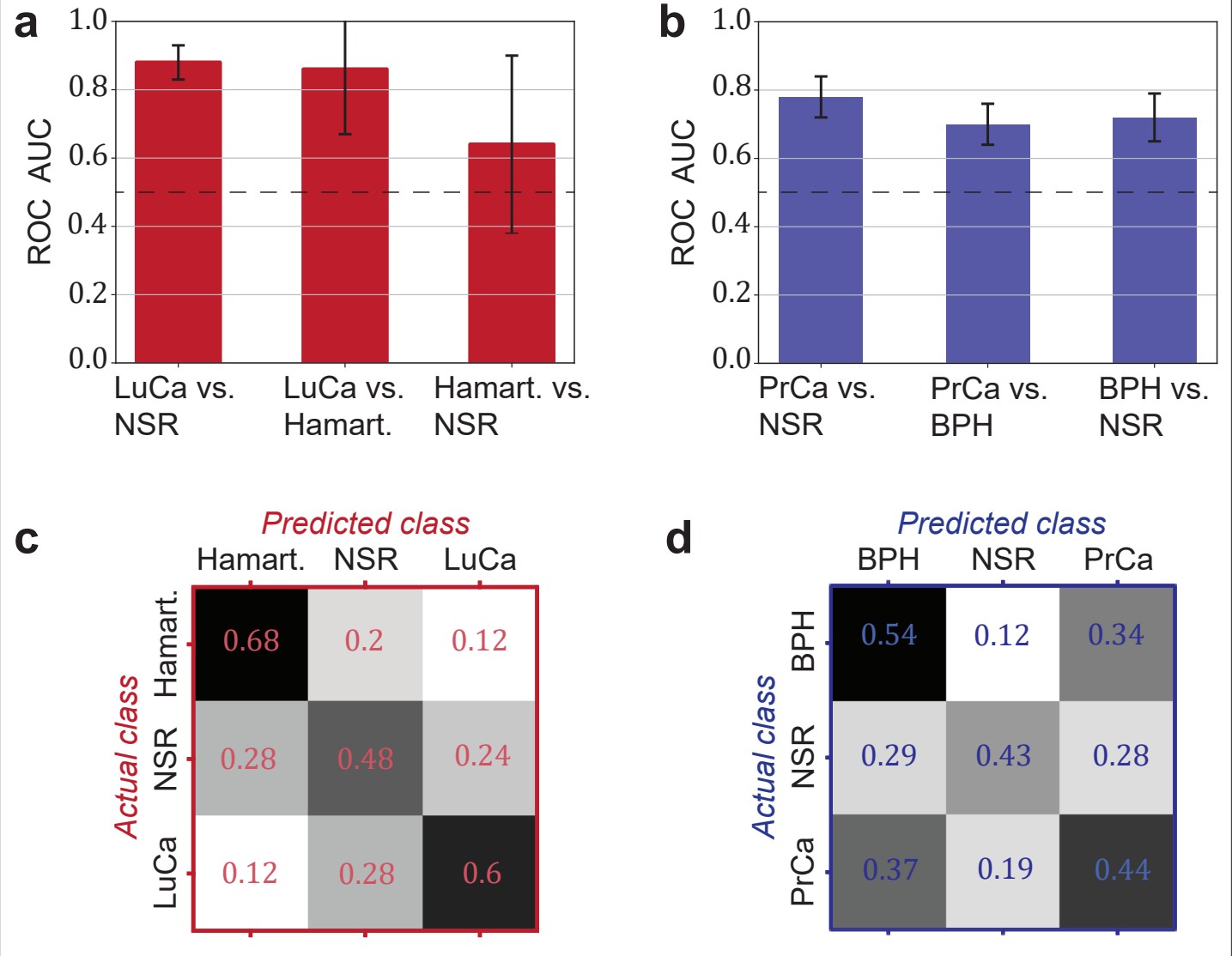

**Figure 4.** Detection efficiency of benign conditions and multiclass classification. (**a**) Pairwise classification performance results between lung cancer (LuCa), hamartoma (Hamart.) and non-symptomatic reference group (NSR) with overall model accuracy of 0.46 ± 0.18, and (**b**) pairwise classification performance between prostate cancer (PrCa), benign prostate hyperplasia (BPH), and NSR with overall model accuracy of 0.43 ± 0.06. The error bars show the standard deviation of the individual results of the cross-validation. Confusion matrix summarizing the per-class accuracies of multiclass classification in (**c**) the LuCa cohort and (**d**) the PrCa cohort. The characteristics of the cohort used for this analysis are given in *Figure 4—source data 1*. Chance level for the three-class classification corresponds to 0.33.

The online version of this article includes the following source data and figure supplement(s) for figure 4:

**Source data 1.** Characteristics of the matched groups utilized for the analysis presented in *Figure 4*.

**Figure supplement 1.** Influence of chronic obstructive pulmonary disease (COPD) in lung cancer (LuCa) detection.

**Figure supplement 1—source data 1.** Characteristics of the matched groups utilized for the analysis presented in *Figure 4—figure supplement 1*.

In contrast, prostate cancer with higher T stage shows neither a significantly better AUC nor a more pronounced differential fingerprint (*Figure 5d*). Instead, the detection efficiency does increase significantly with tumour grade score (*Amin et al., 2017*; *Figure 5e*). A strong correlation between the AUC and Gleason score (*Figure 5f*) could also not be observed.

Finally, the size of the lung cohort allowed us to also investigate the possible effect of metastasis (TNM M1) on the IMFs and their classification performance. As expected from our previous findings, higher AUCs (although not statistically significantly higher) were found in the cohort of locally

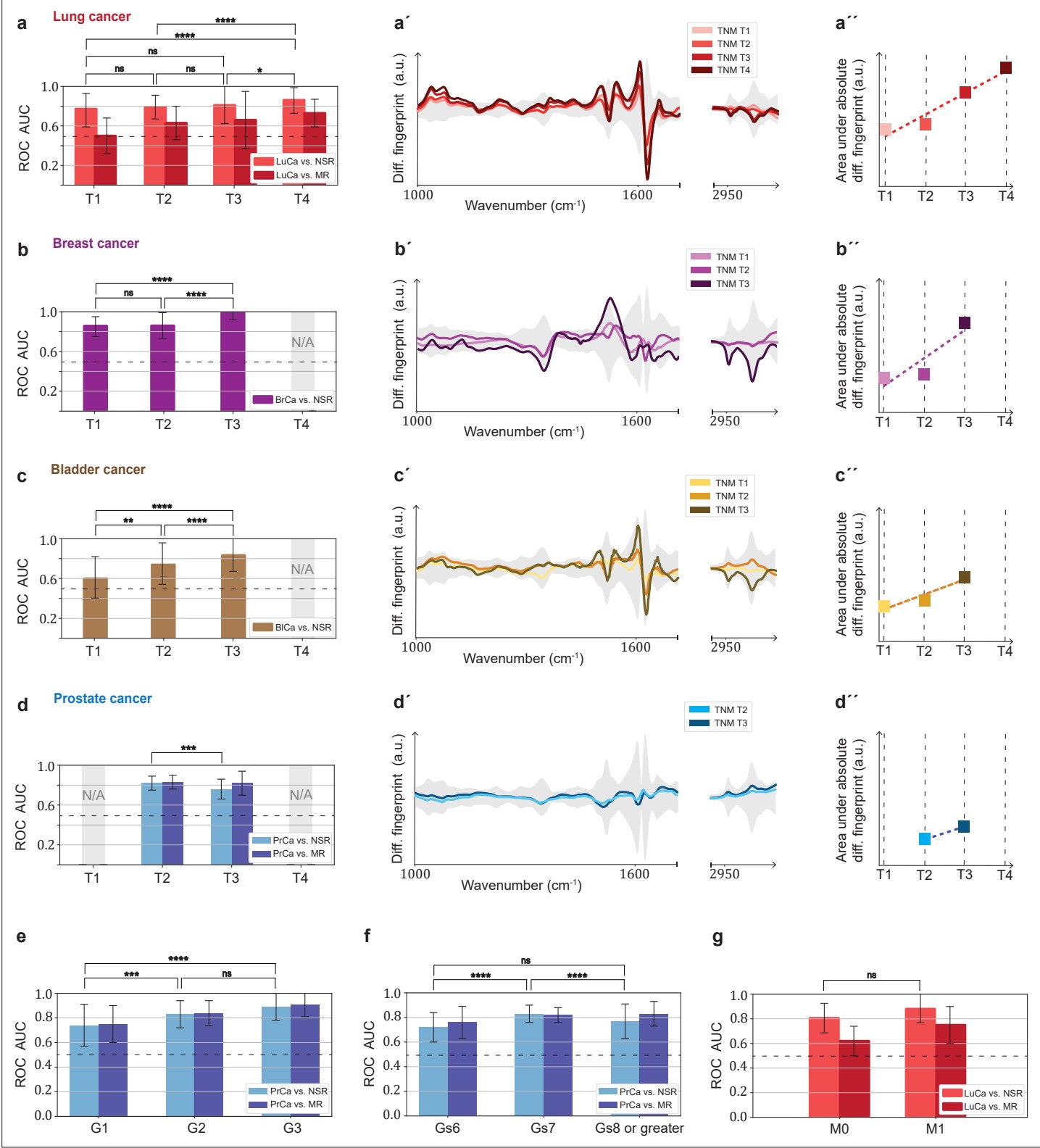

**Figure 5.** Efficiency of binary classification and infrared spectral changes in dependence of tumour progression. (**a–d**) Binary classification performance of lung, breast, bladder, and prostate cancer against references as a function of T-classification (of TNM-staging). (**a'–d'**) Differential fingerprints in relation with the tumour size (TNM class T) for all four cancer entities. (**a''–d''**) Area under the absolute differential fingerprints in relation with the tumour size for all dour cancer entities. The y-axes of the diagrams in the panels (**a'–d'**) and (**a''–d''**) each have the same linear scaling, thus directly comparable. (**e**) Classification performance of prostate cancer versus references as a function of tumour grade score. (**f**) Classification performance of

*Figure 5 continued on next page*

*Figure 5 continued*

prostate cancer as a function of the Gleason score (Gs). (**g**) Classification performance of lung cancer versus references as a function of the metastasis status. The detailed cohort breakdown and classification results are given as *Figure 5—source data 1*, *Figure 5—source data 2*, *Figure 5—source data 3*, *Figure 5—source data 4*. Some cohorts did not include sufficient number of participants so that a reliable machine learning model could not be built and were therefore not evaluated. LuCa: lung cancer; PrCa: prostate cancer; BrCa: breast cancer; BlCa: bladder cancer; NSR: non-symptomatic references; MR: mixed references; n.s.: not significant; *p<10$^{-2}$; **p<10$^{-3}$; ***p<10$^{-4}$; ****p<10$^{-5}$; The error bars show the standard deviation of the individual results of the cross-validation.

The online version of this article includes the following figure supplement(s) for figure 5:

**Source data 1.** Characteristics of the matched groups utilized for the analysis presented in *Figure 5a-d, a'-d' and a''-d''*.

**Source data 2.** Characteristics of the matched groups utilized for the analysis presented in *Figure 5e*.

**Source data 3.** Characteristics of the matched groups utilized for the analysis presented in *Figure 5f*.

**Source data 4.** Characteristics of the matched groups utilized for the analysis presented in *Figure 5e*.

**Figure supplement 1.** Relation between the effect size and the area under the receiver operating characteristic curve (AUC) per wavenumber.

advanced and metastatic lung cancer as compared to cohort of non-metastatic cancer patients only (*Figure 5g*).

Overall, we observe a consistent pattern in agreement with the hypothesis that the signal utilized by the learning algorithm increases with more progressed disease stage (either larger tumour volume, metastatic spread, or tumour grade score). This suggests that the information retrieved from the measured differences between the IMFs of cases and references is connected to tumour-related molecular changes. These changes may be due to larger tumour load leaving a more extensive foot-print on the composition of peripheral blood, or to the fact that tumour progression could have caused a higher systemic response, or to a combination of both. While the correlation between AUC and tumour size was most evident for lung, breast, and bladder cancer, spectral signatures relevant for prostate cancer detection were more strongly connected to the tumour grade score. It is important to note, however, that the observed relation – between the spectrum of the disease and classification efficiency – is not conclusively proven by the current analysis, but only suggested.

## Discussion

We demonstrated the feasibility of blood-based IMF to detect lung, breast, bladder, and prostate cancer with good efficiency. Although previous smaller studies have yielded fairly high classification efficiencies (*Backhaus et al., 2010*; *Elmi et al., 2017*; *Ghimire et al., 2020*; *Medipally et al., 2020*; *Ollesch et al., 2016*; *Ollesch et al., 2014*; *Zelig et al., 2015*), they were either based on low number of participants or might have been affected by confounding factors. Here we provided a rigorous multi-institutional study setup with more than 100 individuals in each case and reference group, 1927 individuals in total, with all case and reference cohorts matched for major confounding factors (n = 1639 individuals upon matching). In addition, we observed that visible infrared spectral signatures correlate with tumour stage, suggesting that the IMFs are significantly affected not only by the presence of tumours but also by the progression of the disease. Furthermore, similar cancer detection efficiencies were achieved with IMFs obtained from blood serum and plasma. This not only confirms the robustness of the results, but also reveals that the method is applicable to both biofluids.

This study provides strong indications that blood-based IMF patterns can be utilized to identify various cancer entities, and therefore provides a foundation for a possible future in vitro diagnostic method. However, IMF-based testing is still at the evaluation stage of essay development, and further steps have to be undertaken to evaluate the clinical utility, reliability, and robustness of the IMF approach (*Ignatiadis et al., 2021*).

First, the machine learning models built within this work will have to be tested with fully independent sample sets. Although the study was designed to account for and minimize the effect of confounding factors, we are aware that these cannot be fully excluded, especially considering that machine learning algorithms are susceptible to them (*Zhao et al., 2020*). To this end, we freeze the current machine learning models, each trained on the data of the entire cohorts of the current study (*Figure 2—source data 2*), and will apply them to a consecutively prospective sample collection to better rule out potential confounders.

Second, it needs to be studied in more detail whether IMFs pick up molecular patterns that are specific to a primary disease process or, more generally, to any secondary inflammatory response. At the current stage, both options seem possible as altered immune responses are also known as primary disease drivers in the context of cancer and may affect genome instability, cancer cell proliferation, anti-apoptotic signalling, angiogenesis, and, last but not least, cancer cell dissemination (*Hanahan and Weinberg, 2011*). This link to systemic effects makes it still difficult to distinguish cancer-specific IMFs (onco-IR-phenotypes) from comorbidities with a strong immune signature (like COPD) in humans in vivo. Nevertheless, we did obtain distinct spectral patterns for all four common cancer entities (*Figure 3b*) indicating different, potentially disease-specific molecular alterations of the IMFs. These changes are likely to be linked to cancer-induced changes since classification accuracy is higher with more advanced cancer stages, which is also reflected in more pronounced differential fingerprints with larger tumour size.

To gain deeper understanding of the specificity of observed spectral changes to the disease patterns studied, it is helpful to investigate their molecular origin. In this context, we do not consider the approach of assigning spectral positions/features to characteristic vibrational modes of functional molecular entities most appropriate. Although widely used in the IR community, due to the very many molecular assignments possible for each spectral position, unambiguous statements of molecular changes herein are not feasible. Instead, a much deeper analysis is required, as recently revealed by *Sangster et al., 2006*; *Voronina et al., 2021*. The latter work involves combination of infrared spectroscopy and quantitative mass spectrometry on the part of the lung cancer sample set used in the current study as well, identifying the molecular origin of the differential infrared fingerprints. These can be partially explained by a characteristic change of proteins that are known to also change due to systemic inflammatory signals. Thereby we highlight the need for further biochemical investigations into the molecular origin of the observed spectral signatures, generally required in the field to address this question conclusively.

In-depth information about the molecular origin of the observed spectral disease patterns will help identify the clinical setting(s) where infrared fingerprinting can make largest contributions to cancer care (e.g. screening, diagnosis, prognosis, treatment monitoring, or surveillance). The specificity of spectral signatures to cancer, along with the obtained sensitivity and specificity in the binary classification (*Table 1*), will influence whether the approach may best complement primary diagnostics, be possibly suited for screening, or even be used for molecular profiling and prognostication. When further validated, blood-based IMFs could aid residing medical challenges: More specifically, it may complement radiological and clinical chemistry examinations prior to invasive tissue biopsies. Given less than 60 µl of sample are required, sample preparation time and effort are negligible, and measurement time is within minutes, the approach may be well suited for high-throughput screening or provide additional information for clinical decision process. Thus, minimally invasive IMF cancer detection could integratively help raise the rate of pre-metastatic cancer detection in clinical testing. However, further detailed research (e.g. as performed for an FTIR-based blood serum test for brain tumour; *Gray et al., 2018*) is needed to identify an appropriate clinical setting in which the proposed test can be used with the greatest benefit (in terms of cost-effectiveness and clinical utility).

Moreover, given the recent evidence of high within-person stability of IMFs over time (*Huber et al., 2021*), serial longitudinal liquid biopsies and infrared fingerprinting of respective samples could eliminate between-person variability by self-referencing and thereby facilitate even more efficient and possibly earlier cancer detection. Once (i) a precise clinical setting is defined and (ii) large-scale, stratified clinical studies controlled for comorbidities can be realized, a systematic, direct comparison to established diagnostics will become feasible and the full potential of infrared fingerprinting can be quantitatively assessed.

For further improvements in the accuracy of the envisioned medical assay, the IR fingerprinting methodology needs to be improved in parallel. Molecular specificity is inherently limited in IR spectroscopy due to the spectral overlap of absorption bands of individual molecules. This might be tackled by chemical pre-fractionation (*Hughes et al., 2014*; *Petrich et al., 2009*; *Voronina et al., 2021*) or by combining IR spectroscopy to methods like liquid chromatography. However, such a pre-fractioning, but also IR fingerprinting itself, would benefit even more from increased spectroscopic sensitivity. Sensitivity of the current commercially available FTIR spectrometer is however limited to detection of highly abundant molecules. Recent developments in infrared spectroscopy demonstrate

the possibility to increase the detectable molecular dynamic range to five orders of magnitude (*Pupeza et al., 2020*) and therefore have the potential to improve the efficiency of infrared fingerprinting.

In summary, infrared fingerprinting reveals the potential for effective detection and distinction of various common cancer types already at its current stage and implementation. Future developments, in terms of instrumentation as well as methodology, have the potential to further improve the detection efficiency. This study presents a general high-throughput and cost-effective framework, and along this, highlights the possibility for extending infrared fingerprinting to other disease entities.

## Methods
### Study design

The objective of this study was to evaluate whether infrared molecular fingerprinting of human blood serum and plasma from patients, reference individuals, and healthy persons has any capacity to detect cancer, specifically targeting detection of four common cancer entities (lung, breast, bladder, and prostate cancer). A statistical power calculation for the sample size was performed prior to the study and is included in the study protocol. Based on preliminary results, it was determined that with a sample size of 200 cases and 200 controls, the detection power in terms of AUC can be estimated within a marginal error of 0.054. Therefore, the aim was to include more than 200 cases for each cancer type. However, upon matching (see also below), it was not always possible to include 200 individuals per group for all analyses of this study. In the analyses where the sample size of 200 individuals per group could not be reached, the uncertainty obtained increased accordingly (as seen in the obtained errors and error bars). The full sample size calculation is available on request from the corresponding authors.

The multi-institutional study on lung, breast, bladder, and prostate cancer also includes subjects with corresponding benign pathologies in the same organs as well as non-symptomatic subjects. Participants provided written informed consent for the study under research study protocol #17-141 and broad consent under research study protocol #17-182, both of which were approved by the Ethics Committee of the Ludwig-Maximillian-University (LMU) of Munich. Our study complies with all relevant ethical regulations and was conducted according to Good Clinical Practice (ICH-GCP) and the principles of the Declaration of Helsinki. The clinical trial is registered (ID DRKS00013217) at the German Clinical Trails Register (DRKS). The following clinical centres were involved in subject recruitment and sample collections of the prospective clinical study: Department of Internal Medicine V for Pneumology, Urology Clinic, Breast Center, Department of Obstetrics and Gynecology, and Comprehensive Cancer Centre Munich (CCLMU), all affiliated with the LMU. The Asklepios Lung Clinic (Gauting), affiliated to the Comprehensive Pneumology Centre (CPC) Munich, and the German Centre for Lung Research, DZL, were further study sites in the Munich region, Germany. In total, blood samples from 1927 individuals were collected and measured (see below). The full breakdown of all participants is listed in *Figure 1—source data 1*.

From the existing dataset, the recorded IMFs were selected for further analysis according to the following criteria:

- Only data from cancer patients with clinically confirmed carcinoma of lung, prostate, bladder, or breast prior to any cancer-related therapy were considered.
- Healthy references were non-symptomatic individuals not suffering from any cancer-related disease nor being under any medical treatment.
- Symptomatic references included patients with COPD or pulmonary hamartoma for lung cancer, and BPH patients for prostate cancer.

From this pre-selected dataset, a further subset was created for each binary classification examined (e.g. lung cancer vs. non-symptomatic references). This selection was done using statistical matching (see below) in such a way that it provides a balanced distribution of gender, age, and BMI. This was to ensure that there is no bias towards any of these factors within the analysis of machine learning. The selection step reduced the number of analysed samples to 1639. It is important to note that given that we have performed evaluations addressing more than one main question, depending on some types of questions, some control samples are appropriately used as matched references for multiple questions.

A full breakdown of all included participants (sample pool) along with the breakdown for each of the investigated binary classification is provided as source data files.

## Statistical matching

Achieving covariate balance between cases and references is an important procedure in observational studies for neutralizing the effect of confounding factors and limiting the bias in the results. In this work, we deploy optimal pair matching using the Mahalanobis distance within propensity score callipers (*Rosenbaum, 2010*). The implementation was done in R (v. 3.5.1). In evaluations where pair matching was not sufficient, optimal matching with multiple references was performed instead.

## Sample collection and storage

Blood samples were collected, processed, and stored according to the same standard operating procedures at each clinical site. Blood draws were all performed using Safety-Multifly needles of at least 21 G (Sarstedt) and collected with 4.9 ml or 7.5 ml serum and plasma Monovettes (Sarstedt). For the blood clotting process to take place, the tubes were stored upright for at least 20 min and then centrifuged at 2000 g for 10 min at 20 °C. The supernatant was carefully aliquoted into 0.5 ml fractions and frozen at –80 °C within 5 hr after collection. Samples were transported to the analysis site on dry ice and again stored at –80 °C until sample preparation.

## Sample preparation and FTIR measurements

In advance of the FTIR measurements, one 0.5 ml aliquot per serum or plasma sample was thawed in a water bath at 4 °C and again centrifuged for 10 min at 2000 g. The supernatant was distributed into the measurement tubes (50 µl per tube) and refrozen at –80 °C. All the FTIR measurements were performed upon two freeze-thaw cycles.

The samples were mostly measured in the order in which they arrived at the measurement site. As sample collection and delivery is to some extent a stochastic process (both cases and references were continuously collected over the entire period), no additional randomization of the measurement order was performed.

The samples were aliquoted and measured in blinded fashion, that is, the person performing the measurements did not know about any clinical information about the samples. The spectroscopic measurements were performed in liquid phase with an automated FTIR device (MIRA-Analyzer, micro-biolytics GmbH) with a flow-through transmission cuvette (CaF$_2$ with ~8 µm path length). The spectra were acquired with a resolution of 4 cm$^{-1}$ in a spectral range between 950 cm$^{-1}$ and 3050 cm$^{-1}$. A water reference spectrum was recorded after each sample measurement to reconstruct the IR absorption spectra. Each measurement sequence usually contained up to 40 samples, resulting in measurement times of up to 3 hr. After each measurement batch, the instrument was carefully cleaned and re-qualified according to the manufacturer's recommendations.

To track experimental errors over extended time periods (*Sangster et al., 2006*), a measurement of quality control serum (pooled human serum, BioWest, Nuaillé, France) was performed after every five samples. The spectra of the QC samples were also used to evaluate the measurement error. We found in a previous study that the measurement error is small when compared to the between-person biological variability of human serum IMFs (*Huber et al., 2021*). A relevant analysis comparing the variability between biological samples and QCs is presented in *Figure 2—figure supplement 1b-b''*. In addition, the results obtained on a subset from plasma and serum samples from the same individuals were similar, indicating that no technical variance or device variation affected the measurement results. Thus, individual samples were not measured as replicates.

## Outlier detection

If an air bubble was present during the measurement, this was immediately noticeable by saturation of the detector. In such cases, the measurement was considered faulty and another aliquot of the sample was measured. After the entire dataset was collected, we performed an additional outlier removal. For this, we used the method of Local Outlier Factor (LOF), as implemented in Scikit-Learn (v. 0.23.2) (*Pedregosa et al., 2011*). LOF is based on k-nearest neighbours and is appropriate for (moderately) high-dimensional data. LOF succeeds in removing samples with spectral anomalies such

as abnormally low absorbance or contamination signatures. Using this procedure, a total of 28 spectra were removed from the dataset.

### Pre-processing of infrared absorption spectra

Negative absorption, which occurs if the liquid sample contains less water than the reference (pure water), was corrected for by a previously described approach (*Yang et al., 2015*). It is known from measurements of dried serum or plasma that there is no significant absorption in the wavenumber region 2000–2300 $cm^{-1}$, resulting in a flat absorption baseline. We used this fact as a criterion for adding to each spectrum a previously measured water absorption spectrum (as provided in *Figure 2—source data 2*) to account for the missing water in the sample measurement and minimize the average slope in this region in order to obtain a flat baseline. All spectra were truncated to 1000–3000 $cm^{-1}$ and the 'silent region,' between 1750 $cm^{-1}$ and 2800 $cm^{-1}$, was removed. Finally, all spectra were normalized using Euclidean ($L_2$) norm. The calculation of the second derivative of the normalized spectra was included in some cases as an additional (optional) pre-processing step.

### Machine learning and classification

To derive classification models, we used Scikit-Learn (*Pedregosa et al., 2011*; v. 0.23.2), an open-source machine learning framework in Python (v.3.7.6). We trained various binary classification as well as multiclass classification models using linear SVM. Performance evaluation was carried out using repeated stratified k-fold cross-validation and its visualization using the notion of the ROC curve for binary problems and the confusion matrix for multiclass classification. The results of the cross-validation are reported in terms of descriptive statistics, that is, the mean value of the resulting AUC distribution and its standard deviation. The calculation of optimal pair of sensitivity and specificity is done by minimizing the distance of the ROC curve to the upper-left corner.

### Statistical analysis

For statistically comparing two groups of spectra (i.e. cases, references), we followed three approaches. First, we calculated the 'differential fingerprint,' defined as the sample mean of the cases minus the sample mean of the reference group. We plot this quantity contrasted against the standard deviation of the reference group for obtaining a visual understanding of which wavenumbers are potentially useful for distinguishing/classifying the two populations. Such a graph serves as a visual representation of what is known as the 'effect size,' which can be obtained by standardizing the differential fingerprint and, as shown in *Figure 5—figure supplement 1*, has an evident relation to the AUC per wavenumber. Secondly, we performed t-test (testing the hypothesis that two populations have equal means) for extracting two-tailed p-values per wavenumber. As a last step, we make use of Mann–Whitney U test (also known as Wilcoxon rank-sum test) for extracting the U statistic and calculating the AUC per wavenumber by the relation $AUC = U/(n1*n2)$, where n1 and n2 are the sizes of the two groups.

## Acknowledgements

We thank Prof. Dr. Gabriele Multhoff, Dr. Stefan Jungblut, Katja Leitner, Dr. Sigrid Auweter, Daniel Meyer, Beate Rank, Sabine Witzens, Christina Mihm, Sabine Eiselen, Tarek Eissa, and Dr. Incinur Zellhuber for their help with this study. In particular, we wish to acknowledge the efforts of many individuals who participated as volunteers in the clinical study reported here. We also thank the Asklepios Biobank for Lung Diseases, member of the German Center for Lung Research (DZL), for providing clinical samples and data.

## Additional information

### Funding

No external funding was received for this work.

### Author contributions

Marinus Huber, Conceptualization, Data curation, Formal analysis, Investigation, Methodology, Validation, Visualization, Writing – original draft, Writing – review and editing; Kosmas V Kepesidis,

Conceptualization, Formal analysis, Investigation, Methodology, Validation, Visualization, Writing – original draft, Writing – review and editing; Liudmila Voronina, Data curation, Formal analysis, Methodology, Writing – review and editing; Frank Fleischmann, Data curation, Investigation, Methodology, Supervision; Ernst Fill, Conceptualization, Formal analysis, Investigation, Methodology; Jacqueline Hermann, Methodology, Project administration, Supervision, Validation; Ina Koch, Data curation, Investigation, Resources, Validation; Katrin Milger-Kneidinger, Resources; Thomas Kolben, Resources, Supervision; Gerald B Schulz, Friedrich Jokisch, Investigation; Jürgen Behr, Nadia Harbeck, Conceptualization, Resources, Supervision, Writing – review and editing; Maximilian Reiser, Christian Stief, Conceptualization, Resources, Writing – review and editing; Ferenc Krausz, Conceptualization, Funding acquisition, Investigation, Methodology, Resources, Supervision, Visualization, Writing – review and editing; Mihaela Zigman, Conceptualization, Formal analysis, Funding acquisition, Investigation, Methodology, Project administration, Resources, Supervision, Visualization, Writing – original draft, Writing – review and editing

### Author ORCIDs
Marinus Huber  http://orcid.org/0000-0001-5309-4475
Kosmas V Kepesidis  http://orcid.org/0000-0002-6391-7743
Ina Koch  http://orcid.org/0000-0002-8766-017X
Mihaela Zigman  http://orcid.org/0000-0001-8306-1922

### Ethics
Clinical trial registration DRKS00013217.
The multi-institutional study on lung, breast, bladder and prostate cancer includes cancer patients as well as subjects with corresponding benign pathologies in the same organs as well as non-symptomatic subjects. Participants provided written informed consent for the study under research study protocol #17-141 and broad consent under research study protocol #17-182, both of which were approved by the Ethics Committee of the Ludwig-Maximillian-University (LMU) of Munich. Our study complies with all relevant ethical regulations, and was conducted according to Good Clinical Practice (ICH-GCP) and the principles of the Declaration of Helsinki. The clinical trial is registered (ID DRKS00013217) at the German Clinical Trails Register (DRKS).

### Decision letter and Author response
Decision letter https://doi.org/10.7554/eLife.68758.sa1
Author response https://doi.org/10.7554/eLife.68758.sa2

---

## Additional files

### Supplementary files
• Transparent reporting form

### Data availability
The datasets analysed within the scope of the study cannot be published publicly due to privacy regulations under the General Data Protection Regulation (EU) 2016/679. The raw data includes clinical data from patients, including textual clinical notes and contain information that could potentially compromise subjects' privacy or consent, and therefore cannot be shared. However, the trained machine learning models for the binary classification of bladder, breast, prostate, and lung cancer are provided within Figure 2—source data 4, along with description and code for importing them in a python script. The custom code used for the production of the results presented in this manuscript is stored in a persistent repository at the Leibniz Supercomputing Center of the Bavarian Academy of Sciences and Humanities (LRZ), located in Garching, Germany. The entire code can only be shared upon reasonable request, as its correct use depends heavily on the settings of the experimental setup and the measuring device and should therefore be clarified with the authors.

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
