## [Decision Letter]

**Acceptance summary:**

This manuscript describes the use of infrared molecular fingerprinting for the detection of multiple types of cancer. The spectral differences between different cancer types have further expanded the diagnostic potential. This work has laid the foundation for future developments in this area.

**Decision letter after peer review:**

Thank you for submitting your article "Infrared molecular fingerprinting of blood-based liquid biopsies for the detection of cancer" for consideration by *eLife*. Your article has been reviewed by 2 peer reviewers, and the evaluation has been overseen by myself as a Reviewing Editor. The reviewers have opted to remain anonymous.

Essential revisions:

1. The paper would benefit from further transparency on the numbers of patients used in the analysis. The authors state that they collected blood (serum and plasma) from 1927 individuals with cancer, symptomatic controls and healthy controls in the abstract. In depth reading and analysis of the paper shows that this dataset is then reduced to 1611 and that plasma samples are only available from a subset of lung and prostate cohorts. They should also explain why 316 patients were removed from the dataset. Further analysis of the supplementary highlights that some discriminations are performed with only 28 samples per disease type for instance the multi cancer discrimination in the female group between bladder, lung and breast and in the male group between bladder lung and prostate there are only 90 per group. As a major claim of the paper is the fact that the authors can distinguish between these types of tumours and that the signatures are significantly different enough a discussion on whether the size of the dataset analysed is sufficient to have that conclusion should be presented. Also the major class within the dataset analysed is the number of subjects with no related conditions (n=635) so essentially the non-symptomatic healthy controls. As the authors in the introduction highlight the fact that there is a need to adequately apply machine learning tools with involvement of a sufficient number of samples and they state that they address this in this paper they should highlight the number of samples involved in each of the analysis they perform or at least discuss the weaknesses that arise. This issue of low sample numbers is an issue in the field and the authors are correct to highlight this in the introduction. The authors state that a power calculation has been performed and is available on request. This should be provided with the paper to show that 28 samples per disease state achieves the required level of significance.

2. The authors utilise an unsupervised PCA based approach to show that there is no difference between the collection sites in supplementary figure 1. However, from experience it is known that a PCA would not show discrimination between the cancers that the authors have analysed. This may be one explanation as to why the authors needed to use an SVM in order to enable the discrimination of the tumour types. To properly enable this conclusion the authors could show the PCA (along with loadings and not just score plots) for the tumour types or perform an SVM on the samples from the three collection sites in order to fully support this conclusion.

3. The authors also state that the pre-processing of the IMFs did not affect the classification result indicating high quality raw data. In order to not mislead a reader is should be noted that what the authors refer to as raw data is not raw data. It is in fact the spectra after a water correction has been performed to correct for negative absorptions. From knowledge the negative absorptions are variable between spectra and it would be interesting for the reader to have a discussion on how often negative absorptions appear in the liquid serum or plasma. Fundamentally the spectra referred to by the authors for subsequent normalisation or 2nd derivatisation have had a correction performed and a such are not raw. In order to fully support this conclusion, it would be interesting to see the impact on the classification accuracy of the raw spectra (the spectra to which the negative absorption has not been applied)

4. Please further expand on what is meant by quality control serum sample – as there a particular serum used for this and what was the procedure for proving quality – do you have a quality test and did any fail at any point?

5. One key objective the study aims to achieve is to improve the specificity of the study by including more suitable control subjects. However, the study still fails to identify which parts of the signal were directly related to the existence of cancer-related molecules. It would be useful for the authors to try to dissect out the most representative cancer signals and to identify the molecules giving rise to those signals.

6. The authors used AUC of ROC curves to reflect the clinical utility of the test. However, for evaluating the usefulness of diagnostic markers, the sensitivities at a specificity of 95% and/or 99% (for screening markers) are frequently used. The presentation of these data would be useful to indicate if the test would be useful in clinical settings.

7. The authors showed that the ROC curves for plasma and serum had similar AUC and concluded that they provided similar infrared information. This point is incorrect. To prove that both plasma and serum can reflect signals from the cancer, the authors need to show that the actual infrared pattern from the plasma and serum are identical.

8. In the discussion, the authors need to discuss:

i. if the current performance of the test is sufficient for clinical use, and how that can assist in the clinical decision process;

ii. how the performance of the test can be improved.

(Merely increasing in the number of test/control subjects is unlikely to lead to a dramatic improvement in the accuracy which makes the test clinically useful.)

*Reviewer #1:*

The authors carried out a multi-center trial to evaluate the potential clinical use of infrared molecular fingerprinting (IMF) for detecting cancer signatures in the blood. In previous proof-of-principle studies, only small numbers of samples were analyzed and that would be subjected to bias related to preanalytical factors and the demographic difference between the cancer patients and control group.

This study performed a one-to-one match of the cancer patients and the controls, and also included symptomatic subjects with benign conditions as controls. This could much better reflect the clinical utility of the test in a real-world situation.

As the study used machine learning to identify the patterns associated with cancer in blood, it did not dissect out which parts of the signal are from the cancer and which are the baseline from the blood cells or other tissue organs. It is expected that the amounts of proteins and DNA from the cancer only constitute a small proportion, most of the signals detected are not from the cancer itself. Hence, this method is still subject to biases related to factors unrelated to cancer, e.g. inflammation. Future studies that involve even larger numbers of controls with a wider variety of benign conditions would be needed to confirm the specificity of the cancer patterns.

The authors used AUC of ROC curves to demonstrate the clinical utility of the test. The sensitivity and specificity of the test are still inadequate in this early version of the test.

This study is a good example of how the evaluation of liquid biopsy tests based on pattern recognition could be performed.

*Reviewer #2:*

The authors set out to achieve a multi institutional study of serum and plasma based IR analysis in order to determine if they could distinguish between breast, bladder, prostate and lung cancer. They have shown an interesting method that can discriminate between disease and symptomatic controls and have provide a good assessment that will surely be useful within the field.

The paper would however benefit from further transparency on the numbers of patients used in the analysis. The authors state that they collected blood (serum and plasma) from 1927 individuals with cancer, symptomatic controls and healthy controls in the abstract. In depth reading and analysis of the paper shows that this dataset is then reduced to 1611 and that plasma samples are only available from a subset of lung and prostate cohorts. They should also publish why there was removal of 316 patients from the dataset. Further analysis of the supplementary highlights that some discriminations are performed with only 28 samples per disease type for instance the multi cancer discrimination in the female group between bladder, lung and breast and in the male group between bladder lung and prostate there are only 90 per group. As a major claim of the paper is the fact that the authors can distinguish between these types of tumours and that the signatures are significantly different enough a discussion on whether the size of the dataset analysed is sufficient to have that conclusion should be presented. Also the major class within the dataset analysed is the number of subjects with no related conditions (n=635) so essentially the non-symptomatic healthy controls. As the authors in the introduction highlight the fact that there is a need to adequately apply machine learning tools with involvement of a sufficient number of samples and they state that they address this in this paper they should highlight the number of samples involved in each of the analysis they perform or at least discuss the weaknesses that arise. This issue of low sample numbers is an issue in the field and the authors are correct to highlight this in the introduction. The authors state that a power calculation has been performed and is available on request. This should be provided with the paper to show that 28 samples per disease state achieves the required level of significance.

The authors utilise an unsupervised PCA based approach to show that there is no difference between the collection sites in supplementary figure 1. However, from experience it is known that a PCA would not show discrimination between the cancers that the authors have analysed. Which is most likely why the authors needed to use an SVM in order to enable the discrimination of the tumour types. To properly enable this conclusion the authors could show the PCA (along with loadings and not just score plots) for the tumour types or perform an SVM on the samples from the three collection sites in order to fully support this conclusion.

The authors also state that the pre-processing of the IMFs did not affect the classification result indicating high quality raw data. In order to not mislead a reader is should be noted that what the authors refer to as raw data is not raw data. It is in fact the spectra after a water correction has been performed to correct for negative absorptions. From knowledge the negative absorptions are variable between spectra and it would be interesting for the reader to have a discussion on how often negative absorptions appear in the liquid serum or plasma. Fundamentally the spectra referred to by the authors for subsequent normalisation or 2nd derivatisation have had a correction performed and a such are not raw. In order to fully support this conclusion, it would be interesting to see the impact on the classification accuracy of the raw spectra (the spectra to which the negative absorption has not been applied)

The strongest aspect of this paper in my opinion is the comparison of lung cancer with common symptomatic diseases such as COPD. There has often been a question within FTIR based clinical spectroscopy of the specificity of the technique with other diseases that are similar in symptomology and that could potentially impact the outcome. The authors have shown this particular discrimination to a substantial level (n = 115 vs 118) which is significant. However it should also be noted that there is a substantial body of work on the use of FTIR on sputum that has shown the ability to differentiate between chronic lung diseases.

[Editors' note: further revisions were suggested prior to acceptance, as described below.]

Thank you for resubmitting your work entitled "Infrared molecular fingerprinting of blood-based liquid biopsies for the detection of cancer" for further consideration by *eLife*. Your revised article has been evaluated by Y M Dennis Lo as Senior Editor and Reviewing Editor, and by the original two reviewers.

As can be seen below, Reviewer #2 has serious concerns about this revised manuscript.

*Reviewer #1:*

The authors have adequately addressed all the concerns I raised in the previous review.

*Reviewer #2:*

I thank the authors for their response, but I now have serious reservations over the paper

The paper in the author’s own words has had the statements weakened and they have removed references to "high-quality" data and as such I do not think it is of sufficient enough novelty and power for *eLife*. In addition there remains confusion over the actual number of samples that have been used.

1. There is still confusion over the numbers used – the authors state 1927, 1611 and 1637 samples. individuals. They only analyse or present 1611 individuals and then when I add up all the numbers in each of the groups in Table 1 supplementary it comes to 2150 so I am clearly confused as to where the samples have come from. I am sorry I my additions are incorrect but really this should be simple and accessible for the reader.

2. The authors have only matched their 200 cases vs 200 cases in 4 out of 64 of the classification they are doing in the entire manuscript and still it seems misleading that they have stated that they have done one. The manuscript overwhelmingly has more non powered classifications than powered ones

3. The authors conclude that the response is related to tumour volume but from looking at figure 5 all of the values are within the error bar of the previous T stage – can the authors comment on this

4. I do not understand why the authors are not including the information on point 2 within the paper – it is not enough to state that it is for review purposes only. This should go in the paper and not be hidden from the public – if the paper is accepted. Do the authors have a reason why they do not want to include this in the paper.

5. In response to point 4 there is quite a spread of the values in the scores plot that authors present, again just for review, the authors need to state the percent variance within this PCA and show the loadings for accurate analysis. Interestingly that this quality control procedure doesn't account for differences between collection sites – as this procedure and the analysis seems to be performed solely at one site and not at the three collection sites this points to an issue in the collection of the samples from different sites that enable to large differences to be observed that is presented in the table on Point 2 – can the authors explain this further – did they have issues with collection differences that are now coming through in the AUC differences between sites only when question by the reviewers?

6. In reference to point 6 in the response to reviews please can the authors comment on the sensitivity values – these are simply stated an don't discussed in the text at all

7. In table 1 why do the authors not compare breast and bladder cancer with SR or MR and instead only state the results for what is reasonably healthy people

8. The authors state if they have 200 versus 200 they have an AUC accuracy within a 0.054 bound on error. Yet at multiple points throughout the paper there error bound is less than 0.054 and they haven't provided the error on most of the classification that are within the 3 x 3 square confusion matrices in the figures

9. From reading this paper I do not think the authors have an independent blind test – the way it was presented first time I didn't pick up on this – please can the authors simply state how they validated the approach. What I would expect is to have a set of patients that is used for train the algorithm and then a completely independent set of patients as a blinded (algorithm blinded at least but hopefully operator blinded) set to prove that the signatures are valid – it seems like the authors have not blind tested the dataset?

Overall the authors have raised some serious concerns on the approach to their methodology by the clarifications provided. They state they have weakened the paper and removed high quality references. They don't want to publish the items performed for review along with the paper and these items raise significant questions about what is the impact of the quality test and even if the collection procedure of the samples is correct and it is also unclear if the classifications have been substantially tested by blinded analysis or if this is simply the results from training the model without testing

---

## [Author Response]

Essential revisions:1. The paper would benefit from further transparency on the numbers of patients used in the analysis. The authors state that they collected blood (serum and plasma) from 1927 individuals with cancer, symptomatic controls and healthy controls in the abstract. In depth reading and analysis of the paper shows that this dataset is then reduced to 1611 and that plasma samples are only available from a subset of lung and prostate cohorts. They should also explain why 316 patients were removed from the dataset.

We very much agree with the referees that it is very important to transparently state the number of individuals enrolled and analysed within the study. Thus, we provide and list the exact numbers and cohort characteristics of each individual analysis provided in the manuscript. This information is provided as source data, linked to each results entity (Figure or Table) of the manuscript.

The referees have correctly stated that although samples from 1927 individuals were measured, only 1611 of them were in depth analysed. The reason for this is our aim to provide only robust answers to the medical questions by performing rigorous matching of diseased and reference individuals such that there were no biases in the cohort characteristics, e.g., due to different ages. For this reason, 316 subjects were not used in the final analyses because they could not be fully properly matched by our rigorous criteria. The exact procedure is described in the Study design sub-section within the Methods section.

To make sure that we are not misleading the readership about the size of our study, we have now modified several statements (in the abstract as well as main text and figure), now even more clearly reflecting that only 1639 individuals were considered for the final in depth analyses (please see page 1 – Abstract; page 4 – line 112; page 8 – line 269; page 9, Figure 1 panel a and d; page 9 – line 279; page 9 – line 289).

Further analysis of the supplementary highlights that some discriminations are performed with only 28 samples per disease type for instance the multi cancer discrimination in the female group between bladder, lung and breast and in the male group between bladder lung and prostate there are only 90 per group. As a major claim of the paper is the fact that the authors can distinguish between these types of tumours and that the signatures are significantly different enough a discussion on whether the size of the dataset analysed is sufficient to have that conclusion should be presented.

Indeed, it is a fact that we observe different spectral signatures for the different tumour types analyzed (Figure 3 a-c). Based on this observation, we were curious to evaluate whether a multi-class distinction is feasible for these cancer entities.

As the reviewers correctly noted, the multi-class classification was performed with a much smaller sample size as compared to the binary classification analyses. This is due to the need to differently match individuals – from all different classes (e.g., we were not able to use the whole group of lung cancer patients, as these were on average differently aged individuals from the prostate cancer group and we had to find matching limits common to all groups of individuals).

Importantly however, although limited in group size, our analysis *suggests* that such a multi-cancer discrimination shall be, in principle, indeed feasible. However, we never intended to claim (based on the available data sets) that we have proven multi-cancer discrimination as feasible, nor was our intention to make it to one of the main claims of the manuscript.

In order to reach a higher transparency to the readership here, we now explicitly point out the smaller data set applied here, along with the correspondingly greater uncertainty of the obtained results (please see page 14 – line 402). In addition, we have newly incorporated a statement noting that although the results indicate the possibility to distinguish between four different types of cancer analysed here, this hypothesis must be further evaluated with significantly larger data sets (please see page 14 – line 405).

Also the major class within the dataset analysed is the number of subjects with no related conditions (n=635) so essentially the non-symptomatic healthy controls. As the authors in the introduction highlight the fact that there is a need to adequately apply machine learning tools with involvement of a sufficient number of samples and they state that they address this in this paper they should highlight the number of samples involved in each of the analysis they perform or at least discuss the weaknesses that arise.

We fully agree with the referees. For this reason, we stated the number of samples involved in each of the analysis that was performed as source data, linked to each results entity.

To further advice the readers to caution in interpretation regarding analyses with small number of samples, we have newly added the following sentence to the power calculation section in the Methods (please see page 5 – starting with line 136):

“Based on preliminary results, it was determined that with a sample size of 200 cases and 200 controls, the detection power in terms of AUC can be estimated within a marginal error of 0.054. Therefore, the aim was to include more than 200 cases for each cancer type. However, upon matching (see also below), it was not always possible to include 200 individuals per group for all analyses of this study. In the analyses where the sample size of 200 individuals per group could not be reached, the uncertainty obtained increased accordingly (as seen in the obtained errors and error bars).”

This issue of low sample numbers is an issue in the field and the authors are correct to highlight this in the introduction. The authors state that a power calculation has been performed and is available on request. This should be provided with the paper to show that 28 samples per disease state achieves the required level of significance.

A statistical power calculation for the binary classifications was indeed performed ahead of time – at the planning stage of this very study prior to any of these analyses. Please find an excerpt of the relevant “Statistical Power Calculations” from the Study Protocol copied here:

“The sample size is determined for estimating the primary endpoint, the AUC, within a pre-specified bound for a 95% confidence interval as well as for a test of the null hypothesis that the AUC = 50% (discrimination for predicting a case versus control no better than flipping a coin) versus the alternative hypothesis that the AUC > 50%, at the 5% significance level and 80% power for specific alternatives. (Hajian-Tilaki K, Journal of Biomedical Informatics 48: 193-204, 2014). The sample size is estimated for each cancer separately.

In preliminary studies of 36 non-small cell lung carcinoma (NSCLC) cancer patients versus 36 controls, mean spectra were separated as shown in (Author response image 1).

**Author response image 1. sa2fig1:** Spectra per wavelength for 36 non-small cell lung cancer patients (NSCLC, red) and 36 controls (healthy, black) with averages in bold.

Assuming an AUC of 0.65 and a 95% confidence interval, a sample of size 200 cases/200 controls can estimate this AUC within a 0.054 bound on error. At the same sample size, a hypothesis test of the null of AUC = 0.50 against the alternative of AUC > 0.50 performed at the 0.05 level would have power > 0.80 to reject the null when the true AUC is 0.58 or greater. The AUC for the spectra value at wavelength 1080 witnessed in Author response image 1 is calculated much greater, at 0.90. This AUC can be estimated with a bound of 0.037 at the planned sample size, and the power for rejecting the null hypothesis for this alternative exceeds 0.90. This AUC estimate is inflated since the wavelength with the highest separation between cases and controls was selected; in practice an automated algorithm integrating all wavelengths together with participant characteristics will be developed with a lower range of AUC in the neighborhood of 0.65 to be expected.”

The most important results of this calculation are also given in the Methods section (please see page 4 – Study design). This calculation shows that an error of 0.054 is expected for a number of 200 cases and 200 control subjects. This is consistent with our presented results of the lung, prostate and breast cancer evaluations (see Table 1). With a correspondingly lower number, the error increases (e.g., for bladder cancer – Table 1). A power calculation for a sample number of 28 was not carried out prior to the study presented here. However, it is also a fact that if one significantly reduces the requirements on the confidence interval, a corresponding power calculation would yield the result that a sample number of 28 would be sufficient. With such a error interval, however, no valid conclusions can be drawn.

Therefore, we made amendments to our manuscript text and now explicitly highlight that the analysis with 28 samples is rather intended as an indication for future investigations. We have now made it very evident in the main text of the manuscript that the results obtained in this study must be independently verified again, with a larger number of samples per group (see our answer to the Point #1 above and Revisions 2-3).

2. The authors utilise an unsupervised PCA based approach to show that there is no difference between the collection sites in supplementary figure 1. However, from experience it is known that a PCA would not show discrimination between the cancers that the authors have analysed. This may be one explanation as to why the authors needed to use an SVM in order to enable the discrimination of the tumour types. To properly enable this conclusion the authors could show the PCA (along with loadings and not just score plots) for the tumour types or perform an SVM on the samples from the three collection sites in order to fully support this conclusion.

We agree with the referees that a PCA-based approach is not the appropriate way to reveal potential bias related to collections sites. We follow the referees’ advice and have for the revision used a supervised SVM-based approach. Specifically, we constructed groups of mixed references (MR), statistically matched to each cancer-entity group, exactly as it was done in the main part of the originally submitted work. But this time, instead of using the entire pool of potential control samples as basis, we perform the matching only on reference samples that come from a specific collection site. We repeated the analysis for all cancer entities and all study sites. The resulting AUCs after the application of SVM in a repeated cross-validation procedure have thus been performed for the review purposes and are given in the table below.

**Author response table 1. sa2table1:** LuCa – lung cancer, BrCa – breast cancer, BlCa – bladder cancer, PrCa – prostate cancer*.*

	LuCa vs.	BrCa vs.	BlCa vs.	PrCa vs.
MR (Study Site 1)	0.96 ± 0.02	0.71 ± 0.07	0.98 ± 0.01	0.77 ± 0.12
MR (Study Site 2)	0.72 ± 0.05	0.88 ± 0.08	0.84 ± 0.09	0.62 ± 0.20
MR (Study Site 3)	0.92 ± 0.03	0.86 ± 0.04	0.77 ± 0.06	0.74 ± 0.04

One typically obtains a different AUC by using different control groups, collected at different sites. However, these variations have various potential causes, including measurement-related effects, differences in sample handling, unobserved differences between the clinical populations recruited at different clinical sites, and of course the size of the training sets used for each evaluation, which can significantly affect the model performance. Although important, it is currently virtually impossible to fully disentangle these effects rigorously. Therefore, the next step in our plans is to independently evaluate the trained models presented in this work (which will also be made publicly available – see Figure 2-source data 4) on a large independent test set, that are currently being collected in frame of ongoing clinical studies. This test set’s design is such that it will not only allow us to evaluate the classification performance in realistic environments, but it will further enable us to assess the effects of covariates (both measurement-related and clinical) using approaches based on generalized linear models (GLM).

Based on this we reformulated and weakened our statement in the main manuscript as follows (please see page 10 – starting with line 318, Revision 4):

“To test whether sample collection, handling, and storage have a potential influence on classification results, we examined data from matched, non-symptomatic, healthy individuals from the three major clinics using principal component analysis (PCA). Considering the first 5 principal components (responsible for 95 % of the explained variance), we could not observe any clustering effect related to data from different clinics (Figure 2—figure supplement 1 and Figure 2-source data 2). However, potential bias due to the above-mentioned influences cannot be fully excluded at the present stage. To this end, samples from different clinical sites are being currently collected to form a large independent test data set, specifically designed to allow us evaluate the effects of clinical covariates – as well as measurement-related ones – relevant for the proposed IMF-based medical assay.”

3. The authors also state that the pre-processing of the IMFs did not affect the classification result indicating high quality raw data. In order to not mislead a reader is should be noted that what the authors refer to as raw data is not raw data. It is in fact the spectra after a water correction has been performed to correct for negative absorptions. From knowledge the negative absorptions are variable between spectra and it would be interesting for the reader to have a discussion on how often negative absorptions appear in the liquid serum or plasma. Fundamentally the spectra referred to by the authors for subsequent normalisation or 2nd derivatisation have had a correction performed and a such are not raw. In order to fully support this conclusion, it would be interesting to see the impact on the classification accuracy of the raw spectra (the spectra to which the negative absorption has not been applied)

We would like to apologize for these unprecise formulations in our initially submitted work regarding this point. The data used for the results shown in Figure 2 d, labelled as raw, meaning that they are indeed water-corrected spectra (please see page 11 – line 348).

We make this point now clear in the revised version as follows (please see page 10 – line 324):

“Furthermore, we investigated the influence of different pre-processing of the IMFs on the classification results, and found that these are not significantly affected by the applied pre-processing (Figure 2 d). Model-diagnostics yielded no signs of overfitting as we added different layers of pre-processing into the pipeline (see Methods for details). Since water corrected and vector normalized spectra typically resulted in slightly higher AUCs but still low overfitting, this pre-processing was kept in all other analyses.”

Moreover, in the revised manuscript, the statement of “high-quality data” has been removed, as it is unnecessary.

4. Please further expand on what is meant by quality control serum sample – as there a particular serum used for this and what was the procedure for proving quality – do you have a quality test and did any fail at any point?

Our strategy to ensure the quality of the measurements as well as analyses is based on two approaches: The first is based on the repeated measurements of pooled human serum of different individuals purchased from the company BioWest, Nuaillé, France (we have added this information to the Manuscript – please see page 6 – line 207). As described in the Method section, we measured the so-called “QC serum” as an internal control after every 5 sample measurements that we performed. The idea behind this is that it is possible to detect potentially relevant drifts of the FTIR device over time.

Importantly however, an unsupervised PCA of the QC measurements performed within 12 months did not reveal major instrument-related drifts in comparison to between-person biological variability.

Please see this in the figure with the PCA graph below that we prepared for the review process:

Secondly, in addition to QC sample measurements, we performed manual and automatic outlier detection and removal at the individual level, with the aim of removing measurement data of samples with spectral anomalies, such as unusually low absorbance or contamination signatures (e.g. from air bubbles). The procedure is described in the Methods section. This way, 28 identified spectra were removed. We added this information to the Methods section (please see page 7 – line 222).

5. One key objective the study aims to achieve is to improve the specificity of the study by including more suitable control subjects. However, the study still fails to identify which parts of the signal were directly related to the existence of cancer-related molecules. It would be useful for the authors to try to dissect out the most representative cancer signals and to identify the molecules giving rise to those signals.

We would like to thank the referees for raising this very important point and for suggesting that we identify cancer-related and potentially specific signatures and the corresponding molecules responsible for these spectral signals. However, in the entire field, hardly any cancer-specific infrared signatures (and the molecules responsible for them) have been identified across several studies so far and the existing work names large molecular classes/groups but no single molecules with actual medical use for diagnostics. The general lack of approaches to unequivocally assign individual molecules to given spectral differences is, in our opinion, the major obstacle to the acceptance and application of infrared fingerprinting for disease detection, in general. Thus, devising a new methodology for identifying molecules behind spectral changes (e.g. mass spectrometry assisted) is beyond the scope of our submitted work.

We would however also like to note that the aim of this very work was not to find out or identify the molecular origin of the spectral signatures provided. In this study, we focused on the question of whether different types of cancer and related diseases have generally different spectral signatures, a problem not addressed in detail previously. Secondly, we evaluated whether these spectral signatures show any correlation with the progression of the disease stages, also not robustly evaluated for these very cancer entities in previously published works (Figure 3 – 5). The question of which changes in molecules are responsible for the observed infrared signatures is a much broader one, and cannot be answered here.

Assigning distinctive spectral features at a certain wavenumber to specific groups of molecules (containing thousands of individual molecules each) may also not be most insightful and useful in a medical perspective (albeit often used within the IR community), especially when considering that assigned vibrational modes occur in many different molecules. Therefore, the observed spectral change can also be caused by a large number and thus variety of different molecules.

In order to target the investigation of the molecular origin of the observed infrared signals, a much deeper analysis is required, as we have recently shown in a very recent publication in Angewandte Chemie – Voronina, L, et al. "Molecular Origin of Blood‐based Infrared Spectroscopic Fingerprints." (2021), https://doi.org/10.1002/anie.202103272. In this work, we have investigated part of the lung cancer sample set here using both FTIR and quantitative mass spectrometry proteomics. The differential infrared fingerprint of lung cancer shown here could be associated with a characteristic change of 12 proteins. The question of whether this change is specific to lung cancer could not be answered this far. Nevertheless, this other Voronina et al. study of ours outlines a strategy on how to attribute the infrared spectral signatures to changes in specific molecular signatures.

Such a complex and detailed analysis as shown in *Voronina et al.* study would exceed the extent of this manuscript. For this reason, we do not attempt to explain the infrared spectral signatures in detail in this very paper and instead rather refer to the mentioned other work of our group (please see page 19 – line 550) and future planned investigations.

6. The authors used AUC of ROC curves to reflect the clinical utility of the test. However, for evaluating the usefulness of diagnostic markers, the sensitivities at a specificity of 95% and/or 99% (for screening markers) are frequently used. The presentation of these data would be useful to indicate if the test would be useful in clinical settings.

We would like to thank the referees for this useful suggestion.

Given the suggestion we have now changed the presentation of our results in this respect and have thus added these values to Table 1 accordingly (please see page 12, most right column of Table 1).

7. The authors showed that the ROC curves for plasma and serum had similar AUC and concluded that they provided similar infrared information. This point is incorrect. To prove that both plasma and serum can reflect signals from the cancer, the authors need to show that the actual infrared pattern from the plasma and serum are identical.

Following the referee’s suggestion, we compared the differential fingerprints (difference of the mean absorbance per wavenumber) for the same comparisons and now incorporate the resulting plots in the Figure 2—figure supplement 2. Even though the two biofluids are subject to noise of different levels (higher for plasma) and have different shapes across the entire spectral range, we do see some resemblance.

Specifically, for lung cancer, where the differential signal carries very distinct and stable pattern, the similarities between infrared signals from serum and plasma are very similar. In the case of differential fingerprints of prostate cancer, however, the comparison is not very conclusive at current stage. The combination of weak signals, high noise and low sample numbers potentially affect the pattern which is based on a simplistic univariate and linear approach that utilizes only a single parameter – the means – of the two distributions.

Summarizing, we agree with the referees that the fact that both ROC curves for plasma and serum are similar in shape and AUC values is a necessary but not a sufficient condition for concluding that both biofluids have similar information content. We therefore decided to reformulate our statement as follows (please see page 11 – line 341):

“Here we compare the diagnostic performance of IMFs from serum and plasma collected from the same individuals for the detection of lung and prostate cancer compared to non-symptomatic and symptomatic organ-specific references. Given that plasma samples were only available for a subset of the lung and prostate cohorts, the results for serum slightly deviate from those presented above due to the different cohort characteristics (Figure 2-source data 3). The detection efficiency based on IMFs from plasma samples was 3% higher in the case of lung cancer and 2% higher in the case of prostate cancer than the same analysis based on IMFs from serum samples. In both cases, the difference in AUC was only of low significance. It is noteworthy that the corresponding ROC curves show similar behaviour (Figure 2—figure supplement 2). These results suggest that either plasma or serum samples could in principle be used for detection of these cancer conditions. However, for carefully assessing whether (i) the same amount of information is contained in both biofluids and (ii) whether this information is encoded in a similar way across the entire spectra, requires yet an additional dedicated study with higher sample numbers.”

8. In the discussion, the authors need to discuss:i. if the current performance of the test is sufficient for clinical use, and how that can assist in the clinical decision process;

We now more explicitly highlight the current state of IMF presented within the study and the future implementation of such an infrared fingerprinting test as a possible in vitro diagnostic assay.

We agree with the reviewers that it is important to provide an estimate, for broader readership, on how far our approach is from actual implementation and applicability within a clinical workflow. To address this, we modified a passage already present in the Discussion section (please see page 19 – line 526) that we are now complementing with the following sentence to make the statement clearer:

“This study provides strong indications that blood-based IMF patterns can be utilized to identify various cancer entities, and therefore provides a foundation for a possible future in vitro diagnostic method. However, IMF-based testing is still at the evaluation stage of essay development, and further steps have to be undertaken to evaluate the clinical utility, reliability, and robustness of the infrared molecular fingerprinting approach (Ignatiadis et al., 2021).“

As we agree that it is required to provide specifics on the possibilities of actual applications, in the revised manuscript we provide an additional statement to the discussion (please see page 19 – starting line 56):

“When further validated, blood-based IMFs could aid residing medical challenges: More specifically, it may complement radiological and clinical chemistry examinations prior to invasive tissue biopsies. Given less than 60 microliters of sample are required, sample preparation time and effort are negligible, and the measurement is performed within minutes, the approach may be well suited for high-throughput screening or provide additional information for clinical decision process. Thus, minimally-invasive IMF cancer detection could integratively help raise the rate of pre-metastatic cancer detection in clinical testing. However, further detailed research (e.g., as performed for an FTIR-based blood serum test for brain tumour (Gray et al., 2018)) is needed to identify an appropriate clinical setting in which the proposed test can be used with the greatest benefit (in terms of cost-effectiveness and clinical utility).”

Altogether, it was not our intention to propose that IMF measurements could be applied as a stand-alone test for medical decision making at the current state of development. It is rather that we envisage the application of the approach as a complementary test that would aid the process of medical diagnostics in time- and cost-efficient manner, once proven for clinical utility.

Not last, since it is early days and the possibilities for technological development of infrared methodology are in front of us are vast, we opt to stay cautious in our statements.

ii. how the performance of the test can be improved.(Merely increasing in the number of test/control subjects is unlikely to lead to a dramatic improvement in the accuracy which makes the test clinically useful.)

As we very much agree with the notion of the referees that the performance of the IR test should be improved, in the originally submitted version of the manuscript we have already dedicated a paragraph to the topic, dealing with suggestions on how to improve molecular sensitivity, specificity as well as dynamic range of molecular concentrations detected.

To further emphasise that these listed suggestions for improvement can also lead to an increased precision of the test, we have further added an additional sentence (please see page 20 – starting with line 586).

“For further improvements in the accuracy of the envisioned medical test, the IR fingerprinting methodology needs to be improved in parallel.”

[Editors' note: further revisions were suggested prior to acceptance, as described below.]

Reviewer #2:I thank the authors for their response, but I now have serious reservations over the paperThe paper in the author’s own words has had the statements weakened and they have removed references to "high-quality" data and as such I do not think it is of sufficient enough novelty and power for eLife. In addition there remains confusion over the actual number of samples that have been used.1. There is still confusion over the numbers used – the authors state 1927, 1611 and 1637 samples. individuals. They only analyse or present 1611 individuals and then when I add up all the numbers in each of the groups in Table 1 supplementary it comes to 2150 so I am clearly confused as to where the samples have come from. I am sorry I my additions are incorrect but really this should be simple and accessible for the reader.

We would like to note that we are sorry to hear that there was still some unclarity. To clear out any possible misunderstanding, we would like to again explain the details of the design of our study:

We collected blood samples from, in total, 1927 individuals (1 sample per individual) from lung, breast, bladder, and prostate cancer patients as well as cancer-free individuals. Please see these numbers as already previously shown in the Figure 1-source data 1 with great detail. After the data collection, we proceeded with the definition of 8 main but separate questions, that we explored with infrared molecular fingerprinting. These are the ones listed in Table 1 of the main text. For each of these questions, we designed a separate case-control study as follows:

1. We selected all collected cases of a particular cancer entity to form the case group;

2. We sub-selected all samples that could be appropriately included as control references based on predefined clinical criteria;

3. We statistically matched (equal number of) controls to cases based on age, gender and body mass index (BMI) – this is the step that contributes to the decrease on the total number of samples used.

The description of this very procedure is provided in the manuscript (see description of the study design and matching in the “Materials and methods” section on pages 18-19). Here, we further clarify its impact on the total number of samples used:

We have collected a large pool of control samples at 3 clinical study sites to be used for addressing any of the 8 independent questions with a well-matched case-control cohort. Given that we had more than one main question, depending on some types of questions, some control samples can be appropriately used as matched references for one question as well as appropriately matched references for a second question as pointed out on page

4. Consequently, our numbers are correct as stated and the accurate number of all samples used is indeed 1639, and is calculated by adding together all samples listed in Figure 1-source data 1 and then subtracting the intersection, i.e. enumerating each sample only once. To further clarify this point, we added the following explanation in the description of the study design (page 18):

“It is important to note that given that we have performed evaluations addressing more than 1 main question, depending on some types of questions, some control samples are appropriately used as matched references for multiple questions.”

We also apologize if in some version of the manuscript the number 1611 appeared. This outdated number corresponded to the total number of samples used before additional classifications (related to supporting materials and required for the peer review) were analysed. Every time new classification questions are defined, clinically-relevant and optimally-matched references are selected anew.

2. The authors have only matched their 200 cases vs 200 cases in 4 out of 64 of the classification they are doing in the entire manuscript and still it seems misleading that they have stated that they have done one. The manuscript overwhelmingly has more non powered classifications than powered ones

We would like to note that the above statement is unfortunately incorrect. The three-step design of case-control groups outlined above has been used for addressing *all classification questions* throughout the manuscript. This includes statistical matching on age, gender and BMI.

The matching is performed following a procedure based on propensity scores, as described in the textbook “Design of Observational Studies” by P. R. Rosenbaum. This type of matching is statistical, with the objective to build groups with similar distributions in terms of age, gender and BMI. In addition, the procedure guarantees that possible small differences, in terms of these parameters, between cases and controls, are non-significant and cannot cause bias. This is checked (within the matching procedure) using logistic regression, showing that a classifier based on age, gender and BMI would yield an AUC of about 0.5.

The main purpose of this study is to answer the main 8 (high-powered) classification questions, as defined in Table 1 and presented in Figure 2. However, due to the richness of the collected cohorts in terms of comorbidities and cancer stages, we decided to investigate even further and thus investigate for possible underlining relationships in the data. Thus, all these further more detailed classification problems and questions, are reported as “extra” results. And for all these results, the associated uncertainty (error bar) was always reported for all evaluations, in a fully transparent way. Low-powered questions yield higher error bars – and this information is just as well provided to the reader, with full transparency.

Clarification**:** In the first part of the description of the study design in the “Materials and methods” section (pages 17-18) we discuss the power calculation that was performed before the collection of the samples and data, as the application to obtain ethical allowance had to be granted to start with the study in the first place. This calculation was performed assuming 200 cases and 200 controls, before the beginning of this study. Its only purpose was to provide us with a theoretical understanding of the amount of error one should expect when analysing a certain number of samples. These numbers (200 cases / 200 controls) are not related to the matching procedure or the actual samples used in the study. This part was introduced in the previous revision after the reviewers requested more information related to the power calculation of the respective study protocol.

3. The authors conclude that the response is related to tumour volume but from looking at figure 5 all of the values are within the error bar of the previous T stage – can the authors comment on this

We wish to highlight that this particular result is not a final conclusion but rather an observation that may well require further corroboration by using larger cohorts. Indeed, we observed a correlation between T stage and classification performance, which prompted us to study this in more detail. The results of this very investigation are presented in Figure 5.

We made it evident to the readers that the distributions of AUC values do overlap between categories, and we did indicate the level of significance for all possible combinations of comparisons presented in the Figure 5.

To get a clearer picture, we also decided to investigate the differences in the fingerprint itself. These results (which are model-independent) are also included within the same Figure 5 and support the pattern observed for the AUCs.

To conclude, with all our findings transparently reported, any reader has all the relevant information required to make any judgments about the existence of an underlining relation. This very relation – between the tumour volume and classification efficiency – is indeed not proven by the current analysis, but only suggested. We are aware that a full proof would require higher amounts of relevant data – and full proof is something that we never claimed. Taken the referee’s point into consideration, we have now expanded on this very point by 1 additional sentence reflecting on these implications (see page 15):

“It is important to note however, that the observed relation – between the spectrum of the disease and classification efficiency – is not conclusively proven by the current analysis, but only suggested. “

4. I do not understand why the authors are not including the information on point 2 within the paper – it is not enough to state that it is for review purposes only. This should go in the paper and not be hidden from the public – if the paper is accepted. Do the authors have a reason why they do not want to include this in the paper.

Our methods and procedures are fully transparent and we did not intend to “hide” any relevant results or information from the scientific public. Full transparency is guaranteed by the entire correspondence during the review process being published along the paper, a policy we are well aware and very much support.

Only because this particular analysis was requested by the reviewers and we didn’t identify any major conclusion that we could have drawn from it, we thought that it would be sufficient to include it into the report only.

However, acknowledging Reviewer #2’s appreciation of the relevance of this result, we have included these data as a supplementary table (see Figure 2 – source data 5) and added the following discussion in the main text (see pages 6 and 7 of the revised manuscript):

“One typically obtains a different AUC by using different control groups, collected at different sites (Figure 2-source data 5). These variations have many potential causes, including measurement-related effects, differences in sample handling, unobserved differences between the clinical populations recruited at different clinical sites, and of course the size of the training sets used for model training, which can significantly affect the model performance. Although important, it is currently not feasible to rigorously disentangle these effects.”

5. In response to point 4 there is quite a spread of the values in the scores plot that authors present, again just for review, the authors need to state the percent variance within this PCA and show the loadings for accurate analysis. Interestingly that this quality control procedure doesn't account for differences between collection sites – as this procedure and the analysis seems to be performed solely at one site and not at the three collection sites this points to an issue in the collection of the samples from different sites that enable to large differences to be observed that is presented in the table on Point 2 – can the authors explain this further – did they have issues with collection differences that are now coming through in the AUC differences between sites only when question by the reviewers?

We would like to point out that in this particular PCA plot, *both* actual samples *and* quality controls (QCs) are plotted. It is actually most reassuring that the spread of real samples is much larger than that of the QCs, providing clear evidence that uncertainties (noise) in sample delivery and fingerprint measurement is much smaller than the natural biological variation existing in the collected samples, which is not present in the case of the QCs.

The two first principal components included in the plot correspond to 93% of the explained variance. Prompted by the reviewer’s comment, we have included these results, along with the loading vectors, in the revised version of Figure 2 —figure supplement 1. We also added a sentence referring to it in the main text (page 19):

“A relevant analysis, comparing the variability between biological samples and QCs is presented in panels b-b’’ of Figure 2 —figure supplement 1.”

The quality controls we use consist of pooled human serum from different people, not related to the study, purchased in large batch. These are not related to the clinical collection sites and therefore they cannot be used for such an analysis. Their main purpose here is to check and control for the reproducibility of the fingerprint measurements (briefly: measurements) and not of the clinical procedures. All measurements took place in our lab (single research facility) and not at the collection sites. We refer to as “collection sites” to the clinics where the blood was drawn from the study participants and not where the actual measurements took place.

We stress that we have not observed any significant differences (i.e. larger than the respective error bars) in the collection of samples among different collection sites. In fact, the same study protocol and same standard operating procedures were used at all sites and all the procedures were fully monitored for correctness along predefined workflows.

The differences in the AUCs can have multiple causes. One of the most probable ones, is the size of the data set used for model training. Machine-learning algorithms typically perform better when trained on larger data sets. This is the reason why we originally had not included this result in the manuscript – we thought that such analysis may be misleading. Prompted by the reviewer’s comment, we have now included these results in the manuscript as explained in the response to the previous comment.

6. In reference to point 6 in the response to reviews please can the authors comment on the sensitivity values – these are simply stated an don't discussed in the text at all

In line with suggestion of the reviewer, we have now included a small paragraph discussing the resulting sensitivities (see page 6):

“Since our approach produces results in terms of continuous variables (disease likelihood) rather than binary outcomes (disease, non-disease), we use the AUC of the ROC as the main performance metric, and thus take advantage of incorporating information across multiple operating points, not limited to a particular clinical scenario. […] For making our results comparable to other studies and possibly to gold standards in cancer detection, we present lists with sensitivity/specificity pairs (see Table 1). In particular, we present the optimal pairs extracted by minimizing the distance between the ROC curve and the upper left corner – a standard practice in studies of this type. In addition, we set the specificity to 95% and present the resulting sensitivities.”

7. In table 1 why do the authors not compare breast and bladder cancer with SR or MR and instead only state the results for what is reasonably healthy people

We would like to note that the reason behind it is that we have no enrolled subjects for such categories and thus no data available in association with benign conditions related to the respective two mentioned organs. Each cancer entity is associated with a different organ and has its own peculiarities. Thus, there is no complete symmetry among all different cancer types. The purpose of this very study is to evaluate whether there is – in principle – *any* possibility to use infrared spectroscopy on hydrated liquid biopsies detect five listed cancer entities. And this is new to the field. However how the proposed approach could be finally integrated in an existing diagnostic workflow and clinical practice is yet an open question that remains to be further addressed in the future.

8. The authors state if they have 200 versus 200 they have an AUC accuracy within a 0.054 bound on error. Yet at multiple points throughout the paper there error bound is less than 0.054 and they haven't provided the error on most of the classification that are within the 3 x 3 square confusion matrices in the figures

We wish to highlight that the estimated error is only a theoretical estimate that was the result of the power calculation that was required and thus included in the Study Protocol submitted for ethics commission approval before the initiation of the study. This part was introduced into the manuscript at the previous revision to fulfil an explicit request of the reviewers.

The classification error bars provided in the manuscript are all calculated based on the results of our cross-validation procedures, i.e. by repeated application of trained models on held-out test sets.

Moreover, we did provide error bars for *all* results of the multi-class classifications and the reviewer is kindly referred to the caption/legend of Figure 3 and Figure 4, where the total accuracy – extracted by the 3x3 confusion matrices – and its error bar are presented.

9. From reading this paper I do not think the authors have an independent blind test – the way it was presented first time I didn't pick up on this – please can the authors simply state how they validated the approach. What I would expect is to have a set of patients that is used for train the algorithm and then a completely independent set of patients as a blinded (algorithm blinded at least but hopefully operator blinded) set to prove that the signatures are valid – it seems like the authors have not blind tested the dataset?

The reviewer is correct that the ideal approach for validation of any new method would be the use of a completely independent set of samples for test purposes. However, this works well only for very large cohorts (> 1000). For small cohorts, of the order of 500 or less (which most of the studies, including ours, fall into), the splitting of the cohort into a “training” set and a set of “left-out” / “test set” data set suffers from a significant degree of arbitrariness, tainting the approach with uncertainties related to the particular choice of the test set. These uncertainties average out for very large (> 1000) cohorts only.

As outlined in the manuscript, we chose to use a *blinded* k-fold cross-validation (CV) to deal with the problem imposed by the arbitrary choice of the test set. CV is a tried and tested, widely-used statistical method to reliably predict/approximate the model’s performance on unseen data. It allows one to judge a model’s performance objectively and to determine if what it has learned is indeed based on actual biological signals, and not random noise.

CV safely and efficiently avoids “cherry picking” of easy-to-predict cases for the test set and thereby overestimating the methods performance in a real clinical setting, by repeatedly splitting the data into k folds. We then go through the folds one-by-one, using the selected fold as the test set and the remaining ones as the train set. In this way, we minimize the risk of choosing an “easy” test set by sheer luck. To get an even more robust estimate, we make use of repeated k-fold CV, which randomly splits the data into a fold multiple times with reshuffled (re-blinded) data. The more repetitions, the more accurate the results. In our case the number of repetitions is 10. In this way, instead of just a single estimate – we obtain a whole distribution of estimates from which we can derive a more thorough understanding of the model’s performance.

To highlight the importance of this issue, earlier this year, the journal Nature Machine Intelligence published a correspondence with Ross King et al. under the title “Cross-validation is safe to use”. The main conclusion was that trusting a left-out test set more than a properly-designed cross-validation procedure is irrational.

Below, we provide a short list of relevant peer-reviewed papers where cross-validation is the method of choice for testing model’s performance:

Skinnider, Michael A., and Leonard J. Foster. "Meta-analysis defines principles for the design and analysis of co-fractionation mass spectrometry experiments." *Nature Methods* (2021): 1-10.

Kim, Hyung Woo, et al. "Dialysis adequacy predictions using a machine learning method." *Scientific reports* 11.1 (2021): 1-7.

Tsalik, Ephraim L., et al. "Host gene expression classifiers diagnose acute respiratory illness etiology." *Science translational medicine* 8.322 (2016): 322ra11-322ra11.

Aguiar, J. A., et al. "Decoding crystallography from high-resolution electron imaging and diffraction datasets with deep learning." *Science advances* 5.10 (2019): eaaw1949.

Peters, Brandilyn A., et al. "Relating the gut metagenome and metatranscriptome to immunotherapy responses in melanoma patients." *Genome medicine* 11.1 (2019): 1-14.